



**Microphysical features of typhoon and non-typhoon rainfall observed in Taiwan, an island in the northwest Pacific.**

**Jayalakshmi Janapati[1], Balaji Kumar Seela[1], Pay-Liam Lin[1,2, 3*], Meng-Tze Lee [4], Everette Joseph[5]**

[1]Institute of Atmospheric Physics, Department of Atmospheric Sciences, National Central University, Zhongli district, Taoyuan city, Taiwan

[2]Earthquake-Disaster & Risk Evaluation and Management Center, National Central University, Zhongli district, Taoyuan city, Taiwan.

[3]Research Center for Hazard Mitigation and Prevention, National Central University, Zhongli district, Taoyuan City, Taiwan

[4]Department of Atmospheric and Oceanic Sciences, McGill University, Montreal, Quebec, Canada

[5]National Center for Atmospheric Research, Boulder, Colorado

**\*Correspondence to:**

Prof. Pay-Liam Lin

Institute of Atmospheric Physics, Department of Atmospheric Sciences

National Central University, Zhongli district, Taoyuan City, Taiwan

Phone: 03-422-3294 03-422-7151 ext. 65509

E-mail: tliam@pblap.atm.ncu.edu.tw





**Abstract.**
The microphysical features of the typhoon (TY) and non-typhoon (NTY) rainfall in
summer seasons are analyzed using long-term (2004 to 2016) data from the impact disdrometer
installed in north Taiwan. The RSD stratified based on rainfall rate showed distinct RSD
characteristics between TY and NTY rainfall. More (less) number of small (big) size raindrops
are noticed in TY rainfall than NTY rainfall. RSD features in terms of gamma parameters are
studied for these two weather regimes. The mass-weighted mean diameter ($D_m$) values are higher
in NTY than TY rainfall, and an inverse behavior is observed for the normalized intercept
parameter ($N_w$). Even after separating the rainfall regimes into convective and stratiform type, a
large $D_m$ is found in NTY compared to TY precipitation. Distinct variations in $Z$ –$R$, $D_m$–$R$, $N_w$–
$R$, $KE$–$R$, and $KE$–$D_m$ relations are noticed between TY and NTY rainfall.  Possible mechanisms
responsible for the RSD variations between TY and NTY are discussed using reanalysis, remote-
sensing, and ground-based radar datasets.
**Keywords:** typhoon, non-typhoon, disdrometer, rainfall kinetic energy,  north Taiwan



## 1. Introduction

Taiwan is an island in the northwest Pacific, with its central mountain range (average
height of around 2 km and peaks of ~ 4 km) extending from south to north direction. It is
surrounded by the East China Sea in the north, the Philippine Sea in the east, Luzhon strait in the
south, and the South China Sea in the southwest. This island is influenced by two major
prevailing monsoon regimes: southwesterly monsoon that occur from May to August, and
northeasterly monsoon spanning from September to April (Chen and Chen, 2003). Further,
rainfall in Taiwan is categorized into winter (December to February), spring (March to April),
mei-yu (mid-May to mid-June), summer (mid-June to August), typhoon (May to October), and
autumn (September to November) regimes (Chen and Chen, 2003). Among the above mentioned
seasons in Taiwan, the summer season is associated with thunderstorm and typhoon weather
systems with intense precipitation than the other seasons. Over this island, though there were
numerous studies on rainfall characteristics of different weather systems in different seasons
(Chen et al., 1999;Chen et al., 2007;Chen et al., 2010;Chen and Chen, 2011;Liang et al., 2017;Tu
and Chou, 2013), there are few attempts in elucidating the microphysical aspects of precipitating
clouds, especially the raindrop size distribution (RSD).
The RSD offers applications in diverse fields like meteorology, hydrology, remote
sensing, and provide an insight into the precipitation microphysics (Rosenfeld and Ulbrich,
2003). The RSD is useful in designing the rainfall estimation algorithms for radar measurements
(Ryzhkov and Zrnić, 1995), improving the cloud modeling parameterization (McFarquhar et al.,
2015), assessing rainfall erosivity relations (Janapati et al., 2019), validating the remote sensing
instruments (Liao et al., 2014;Nakamura and Iguchi, 2007), and in rain attenuation studies (Chen



et al., 2011). Owing to RSD applications mentioned above, there were RSD reports for spatial,
seasonal (Thompson et al., 2015;Jayalakshmi and Reddy, 2014;Seela et al., 2017;Seela et al.,
2018;Krishna et al., 2016) variations,  storm to storm, within the storm (Kumari et al.,
2014;Maki et al., 2001;Jung et al., 2012), and in different precipitations (Tokay and Short,
1996;Krishna et al., 2016).

There has been an increasing interest in the RSD studies to elucidate the hydrological

(Lin and Chen, 2012;Lu et al., 2008;Janapati et al., 2019;Chang et al., 2017), and microphysical
characteristics (Chu and Su, 2008;Jung et al., 2012;Seela et al., 2017;Seela et al., 2018;Lee et al.,
2019;Janapati et al., 2020) of different precipitating clouds in Taiwan. For instance, Chu and Su
(2008) investigated the slope-shape relations for four different precipitations in north Taiwan. In
the southern part of Taiwan, the microphysical features of a convective system (a squall line)
were investigated by  Jung et al. (2012), and they noticed larger  $D_m$ values ($N_w$, μ, and Λ) in
convective precipitation than the maritime clusters. In north Taiwan, the RSD of thirteen landfall
typhoons were reported by  Chang et al. (2009). Spatial variations of RSD by Seela et al. (2017)
showed that the summer seasons rainfall in Taiwan has more large drops when compared to
Palau. They demonstrated that terrain influenced deeply extended convective clouds with more
aerosol loading in Taiwan are responsible for the RSD variations between these two islands.
Seasonal characteristics of RSD by Seela et al. (2018) established that deep convective clouds in
summer and warm clouds in winter seasons resulted in higher $D_m$ values in summer than winter
seasons. Microphysics of seasonal rainfall in north Taiwan were analyzed by Lee et al. (2019),
and they perceived the highest mean $D_m$ values in the summer and highest concentration
($\log_{10}N_w$) in the winter. Recently, disdrometer observations in Taiwan and India sites showed

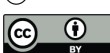



higher $D_m$ values in Pacific Ocean tropical cyclones than the Indian Ocean tropical cyclones
(Janapati et al., 2020).

There were studies in elucidating the tropical cyclones and non-tropical cyclones RSD
characteristics of a given season over India, Australia, China, and Japan (Radhakrishna and
Narayana Rao, 2010;Kumar and Reddy, 2013;Deo and Walsh, 2016;Chen et al., 2019;Chen et
al., 2017). At a south India station, Gadanki, more small and mid-size drops were observed in
tropical cyclonic rainfall than non-cyclonic rainfall (Radhakrishna and Narayana Rao, 2010). At
another south India station, Kadapa, more large drops were noticed in northeast monsoon
thunderstorm precipitation than the tropical cyclone rainfall (Kumar and Reddy, 2013). In
Australia, Deo and Walsh (2016) illustrated the tropical cyclones and non-tropical cyclones
RSDs and demonstrated higher $D_m$ values in non-tropical cyclones than tropical cyclones
rainfall. The polarimetric radar variables computed with the 2DVD for the typhoon, Mei-yu, and
squall line precipitations over Easter China showed distinct differences among these
precipitation types (Chen et al., 2017). Over south China, distinct differences in rain integral
parameters of typhoons and squall lines were noticed by Zhang et al. (2019). They concluded
that it is crucial to adopt the precipitation specific rainfall estimators. Recently, Chen et al.
(2019) examined the typhoon and mei-yu rainfall's RSD characteristics and noticed maritime
behaviors in the typhoon rainfall and continental behavior in mei-yu rainfall.

Even though there were reports on the rainfall characteristics of the typhoon and non-
typhoon rainfall over Taiwan (Chen and Chen, 2011;Tu and Chou, 2013), there is a lack in
identifying the  RSD features of the typhoon and non-typhoon weather systems associated with





summer seasons over Taiwan. Hence, the current study is motived with the below-mentioned
objectives: Do the RSD features of the typhoon and non-typhoon rainfall events in summer
seasons for Taiwan show similar or dissimilar characteristics? And if they exist, what are the
possible reasons for the RSD differences. Do the typhoon and non-typhoon RSD parameters
display comparable/different features to the previous studies? Can we adopt default or tropical
rainfall estimation ($Z$–$R$) relations, or do we need to revise them? Henceforth, an attempt is made
to study the typhoon and non-typhoon RSD characteristics in summer seasons (16th June to 31st
August from 2004 to 2016) at the north Taiwan site.

**2.  Data sets used**

The current study utilized the Joss–Waldvogel disdrometer (JWD) (Joss and Waldvogel,

1969) measurements in NCU (24$^o$ 58' N, 121$^o$ 10' E), Taiwan, from 16 June – 31 August
(summer in Taiwan) for 2004 to 2016 years. The summer seasons rainy days in north Taiwan are
separated into typhoon (TY) and non-typhoon (NTY) regimes. In identifying the rainfall
amounts of typhoons over Taiwan, previous studies adopted different criteria (Tu and Chou,
2013;Chu et al., 2007;Chen et al., 2010). For instance, if a typhoon enters a rectangular box of
21$^o$-26$^o$ N and 119$^o$-125$^o$ E, then the corresponding rainfall over Taiwan was considered as
typhoon rainfall by Chu et al. (2007).  Chen et al. (2010) used a rectangular box of 19.5$^o$-27.5$^o$
and 117.5$^o$-124.5$^o$ E to define as the typhoon rainfall when a typhoon invades this box.
Similarly, Tu and Chou (2013) used another grid box of 18$^o$-29.5$^o$ N and 116$^o$-126$^o$ E to define
the typhoon rainfall.  In the present study, the rainfall at the disdrometer site is considered as
typhoon-induced rain if the typhoons center is $\leq$ 500 km from the disdrometer (Janapati et al.,
2019). The rest of the rainy days that were not classified as the typhoon-induced storm were



categorized as non-typhoon rainfall. With this criteria, a total number 59 typhoon, and 131 non-
typhoon rainy days were recorded by the JWD in NCU from 2004 to 2016 (excluding 2008 and
2009 years). The geographical location of Taiwan with the disdrometer site (indicated with green
color circle) is shown in Fig. 1.

The rain/RSD parameters like raindrop concentration $N(D)$ ($mm^{-1}$ $m^{-3}$), radar reflectivity

factor $Z$ ($mm^6$ $m^{-3}$), liquid water content $W$ (g $m^{-3}$), rainfall rate $R$ (mm $h^{-1}$), total number
concentration $N_t$ ($m^{-3}$),  normalized intercept parameter, $N_w$ ($m^{-3}$ $mm^{-1}$), shape parameter $\mu$ (-),
and slope parameter $\Lambda$ ($mm^{-1}$), and mass-weighted mean diameter $D_m$ are estimated from the
JWD measurements. The formulations for these rain/RSD parameters are detailed in (Seela et al.,
2017;Seela et al., 2018;Tokay et al., 2001;Bringi et al., 2003;Tokay and Short, 1996).

In addition to rain parameters, the rainfall kinetic energy ($KE$) that is expressed in $KE$

flux ($KE_{time}$, in J $m^{-2}$ $h^{-1}$) and $KE$ content ($KE_{mm}$, J $m^{-2}$ $mm^{-1}$) are computed for TY and NTY
rainy days of summer seasons for north Taiwan by following the procedures of (Fornis et al.,
2005;Salles et al., 2002;van Dijk et al., 2002).

Along with the disdrometer data, remote-sensing (TRMM and MODIS) and reanalysis

(ERA-interim) data sets are used to elucidate the typhoon and non-typhoon rainy days'
microphysical characteristics. Bright band and storm heights from TRMM satellite data product
(2A23)(Iguchi et al., 2000;Kummerow et al., 2001), Cloud effective radii of liquid and ice
particles from MODIS satellite data product (MOD08_D3) (Platnick et al., 2015;Remer et al.,
2005;Nakajima and King, 1989), water vapor, convective available potential energy, relative





humidity and temperature profiles from ERA-Interim (Dee et al., 2011)  are considered for TY
and NTY rainy days. Both remote-sensing and reanalysis data sets are interpolated to 0.125° ×
0.125° over the disdrometer site.  A brief description of these data sets can be found in (Seela et
al., 2018;Janapati et al., 2020).

Besides remote-sensing and re-analysis data sets, the radar reflectivity profiles from
radars mosaic are used to reveal the rainfall characteristics of TY and NTY rainy days. The Z
profiles are obtained from the six radars that are depicted with red triangles in Fig. 1. The Z
profiles for the period of 2005-2014 are used over the observational sites, and an explanation of
the reflectivity profiles from six ground-based radars is provided in Seela et al. (2018).

**3.  Observational Results**
The disdrometer rainy days, which have nearly equal amounts of rainfall to that of the
collocated rain gauge observations (data points in Fig. 2), are used to demonstrate the raindrop
size distribution (RSD) features of the typhoon (TY) and non-typhoon (NTY) rainfall. The JWD
recorded a total of 23074 and 20368 rainy minutes for TY and NTY precipitations. The mean
raindrop concentrations for TY and NTY rainfalls are provided in Fig. 3a. Throughout this paper,
raindrops of diameter greater than 3 mm, 1−3 mm, and less than 1 mm are named, respectively,
as large, mid, and small size drops (Tokay et al., 2008;Seela et al., 2018). From Fig. 3a, it can be
noticed that the NTY rainy days have more large size drops than the TY rainy days. A clear
separation between TY and NTY rainfall RSD can be noticed. Because of drop concentrations
dependency on the rainfall rate, it is difficult to interpret the RSD difference between TY and
NTY rainfall from Fig. 3a. Hence we adopted the normalization (Testud et al., 2001) to the TY



and NTY rainy days RSD. This method is independent of the shape of the observed raindrop
spectra. It can be useful in comparing the RSD of different precipitations and inspecting them for
indications of differences in the physical processes producing the rain.  The drop diameter ($D$,
mm), raindrop concentrations [$N(D)$, mm$^{-1}$ m$^{-3}$] of TY and NTY precipitations are normalized
by mass-weighted mean diameter ($D_m$, mm) and normalized intercept parameter ($N_w$, mm$^{-1}$ m$^{-3}$),
respectively, and are shown in Fig. 3b.  From the figure, it is apparent that the normalized NTY
spectra depart noticeably from the TY spectra for $D/D_m > 2$, suggesting that the rain production
in TY and NTY is involved with substantially different microphysical processes.

The probability distribution functions (PDF) of $D_m$ (mass-weighted mean diameter in

mm), $\log_{10}N_w$ ($N_w$ is normalized intercept parameter in mm$^{-1}$ m$^{-3}$), $\log_{10}R$ ($R$ is rainfall rate in
mm h$^{-1}$), and $\log_{10}W$ ($W$ is the liquid water content in g m$^{-3}$), for TY and NTY rainy days, are
depicted in Fig. 4. The PDF distributions of $D_m$ in TY and NTY rainy days clearly show that
NTY rainy days have higher distributions than TY rainy days for $D_m$ values great than 1.7 mm
(Fig. 4a). The normalized intercept parameter,$\log_{10}N_w$ ($\underline{N_w}$ in m$^{-3}$ mm$^{-1}$) PDF distributions show
peak values around 3.7 m$^{-3}$ mm$^{-1}$ and 3.4 m$^{-3}$ mm$^{-1}$, respectively, for TY and NTY rainy days
(Fig. 4b). The TY and NTY rainy days have peak PDF distributions of $\log_{10}R$ around 0.3 and 0,
respectively (Fig.4c). The PDF of $\log_{10}W$ shows a higher percentage at lower $\log_{10}W$ values
($\log_{10}W < -1$) in NTY rainy days, and a higher percentage at higher $\log_{10}W$ values ($\log_{10}W > -1$)
in TY rainy days (Fig. 4d). Further, a statistical (Student's t-test) test performed for parameters in
Fig. 4 showed that the results rejected the null hypothesis at significance levels of 0.05 and 0.01.






### 3.1 Contribution of raindrop diameters to $N_t$ and $R$


The contribution of raindrop diameter classes (diameter < 1 mm, 1−2 mm, 2−3 mm, 3−4
mm, and 4−5 mm) to $N_t$ (m$^{-3}$) and $R$ (mm h$^{-1}$) for TY and NTY rainy days are shown in Fig. 5.
From Fig.5a & b, it can be seen that for both TY and NTY rainy days, with the increase of drop
diameter classes, contribution to total number concentration decreased, while that of rainfall rate
increased and then decreased. This characteristic also agrees with the findings of previous studies
on tropical cyclones (Chen et al., 2019) and summer season rainfall (Wu et al., 2019). For both
TY and NTY rainy days, small size drops (< 1 mm) predominantly contributed to a large number
concentration (> 70%) and about 10% to rainfall rate. Raindrops with diameter 1−2 mm
contributed to number concentration around 20% for TY and NTY rainy days and to rainfall rate
around 60% (55%) for TY (NTY) rainy days. The contribution of drops with diameters 2−3 mm
to number concentration is negligible (for both TY and NTY days), and the rainfall rate is above
20% for TY and NTY rainy days. Fig. 5a and b clearly emphasize that small (< 1 mm) and mid-
size drops (1−3 mm) contributed to a higher percentage of total number concentration and
rainfall rate.

The ratio of $N_t$ (m$^{-3}$) occurrence in TY and NTY rainy day to the both (TY+NTY) rainy
days in each drop diameter class are illustrated in Fig.5c. Similarly, the ratio of $R$ (mm h$^{-1}$)
occurrence in TY and NTY rainy day to the both (TY+NTY) rainy days in each drop diameter
class are illustrated in Fig.5d. The percentage of $N_t$ (m$^{-3}$) in the first three drop diameter classes
(< 1 mm, 1−2 mm, 2−3 mm), i.e., for the small and mid-size drop classes, is higher in TY than
NTY rainy days. On the other hand, for the raindrop diameter classes greater than 3 mm, the
occurrence percentage of $N_t$ (m$^{-3}$) is higher in NTY than TY rainy days. Similar to the $N_t$ (m$^{-3}$),





the percentage of rainfall rate is higher in TY than NTY rainy days for small and mid-size drops,
and an opposite feature can be seen for large drops (> 3 mm).

**3.2 Segregation of RSD based on rainfall rates**
The total RSDs of TY and NTY rainfall are stratified into seven rainfall rate classes.
These rainfall rate classes are considered with the below-mentioned conditions. There were
sufficient data points in every class for TY and NTY rainy days, and the mean values of rainfall
rates in each category are nearly equal. A similar classification criterion was assumed by
(Jayalakshmi and Reddy, 2014;Deo and Walsh, 2016;Seela et al., 2017).  Statistical values of
these seven rainfall rate classes in TY and NTY rainy days are given in Table 1. Each rainfall
rate class's mean value is nearly equal in TY and NTY rainy days. Except for fourth and fifth
rainfall rate classes (C4 and C5), the skewness is higher in NTY than TY rainy days, and both
the weather systems (TY, NTY) showed +ve skewness, which indicates that the rainfall rates are
focused on the left to the mean.

Following the procedure mentioned in Seela et al. (2018), the RSD variations between
TY and NTY rainfall in seven rainfall rate classes are evaluated in terms of drop concentration
percentage and are illustrated in Fig. 6. The drop-concentration percentage is the ratio of $N(D)$ in
TY or NTY rainy days for  the raindrop diameter $D$ and rainfall rate class $R$ to the raindrop
concentration accumulations in TY and NTY rainy days. The percentage contribution of $N(D)$
for TY and NTY rainy days demonstrates that, for all the rainfall rate classes, the small and mid-
size drops (< 3 mm) have a higher percentage in TY than NTY rainy days. Whereas, the large
drops (> 3 mm) show a higher percentage of $N(D)$ for NTY than TY rainy days.





Distributions of $D_m$ (mm) and $\log_{10}N_w$ (m$^{-3}$ mm$^{-1}$) for seven rainfall rate classes are
depicted with box plots in Fig. 7. From Fig. 7a, it is evident that with rainfall rate class increase,
an increase in $D_m$ values can be seen for both TY and NTY rainy days, which is due to a raise in
large size drops concentration and a reduction in the concentration of small drops with the
rainfall rates increase (Rosenfeld and Ulbrich, 2003;Krishna et al., 2016). On the other hand, $D_m$
values are higher in NTY than TY rainy days in all the rainfall rate classes due to the more
concentration of mid-size and small size raindrops in TY than NTY rainy days (Fig.6).
Compared to $D_m$, for all seven rainfall rate classes, the $\log_{10}N_w$ values are higher in TY than NTY
rainy days (Fig.7b).

Distributions of $D_o$ [$D_o = (3.67+ \mu)/\Lambda$] and $\log_{10}N_w$ in different rainfall rates classes ($< 5$,
$5-10$, $10-30$, $30-50$, and $> 50$ mm h$^{-1}$) for TY and NTY rainy days are illustrated in Fig. 8a & b.
The stratiform and convective classification lines of Thompson et al. (2015)  and Bringi et al.
(2009) are denoted, respectively, with horizontal black dotted and slant solid lines. With the
rainfall rate classes increase, the distributions $D_o$, and $\log_{10}N_w$ are getting narrowed for both TY
and NTY rainy days. The $D_o$ and $\log_{10}N_w$ data points were distributed in convective and
stratiform regions of Bringi et al. (2009) precipitation classifications line (inclined solid line in
Fig. 8) for rainfall rates $> 10$ mm h$^{-1}$ and $< 10$ mm h$^{-1}$, respectively. Mean values of $D_o$ and
$\log_{10}N_w$ in different rainfall rates for TY and NTY rainy days are depicted in Fig. 8c & d. For
both TY and NTY rainy days, mean $D_o$ values increase with the increase of $R$ classes. Moreover,
for $R > 10$ mm h$^{-1}$, mean $D_o$, and $\log_{10}N_w$ values were distributed in the convective region of
Bringi et al. (2009) classification line (Fig. 8c & d). Further, the TY rainy days mean $\log_{10}N_w$
values are found to be equal or very slightly higher than the Thompson et al. (2015) rainfall

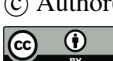



classification line for rainfall rates > 10 mm h$^{-1}$ (Fig. 8c). On the other hand, for all rainfall rate
classes, the NTY rainy days, mean log$_{10}N_w$ values are smaller than the rain classification line of
Thompson et al. (2015) (Fig.8d). This shows that, in separating the TY and NTY rainy days of
summer seasons over north Taiwan into stratiform and convective type, Bringi et al. (2009)
classification method is superior to that of the Thompson et al. (2015).

**3.3 RSDs in precipitation types**
The RSDs were found to show distinct features with the precipitation types, and there
were numerous reports in separating the precipitations into stratiform and convective types (Ma
et al., 2019;Jayalakshmi and Reddy, 2014;Ulbrich and Atlas, 2007). For instance, Tokay and
Short (1996) reported variations in convective precipitations to that of the stratiform regimes.
There were reports in emphasizing to adopt precipitation specific rainfall estimation relations
(Ulbrich and Atlas, 2007). In this work, TY and NTY rainfall are segregated to convective and
stratiform regimes using the rain classification method detailed in Ma et al. (2019). Distributions
of mean $N(D)$ (m$^{-3}$ mm$^{-1}$) with raindrop diameters for TY and NTY rainy days in two
precipitation regimes are depicted in Fig. 9a. The drop concentration of the convective rainfall is
higher than the stratiform for all the drop diameters except for the first drop size bin. Concave
shape with broader distributions of $N(D)$ in convective than stratiform is due to the breakup of
large drops by collisions (Hu and Srivastava, 1995). Present RSD features in both precipitation
types are similar to the earlier studies for continental (Jayalakshmi and Reddy, 2014) and oceanic
regions (Krishna et al., 2016). On the other hand, in stratiform and convective regimes, the mid-
size and large drops concentration is higher in NTY than TY rainy days. Variations in $D_m$ and
log$_{10}N_w$ for both precipitations of TY and NTY are depicted in Figure 9b. The maritime and





continental convective clusters of Bringi et al. (2003) are depicted with gray rectangles. For both
TY and NTY rainy days, larger mean $D_m$ and $\log_{10}N_w$ can be seen for convective precipitation. In
contrast to that, in stratiform and convective regimes, the NTY rainy days have smaller $\log_{10}N_w$
and larger $D_m$ values than TY rainy days.

**3.4 Rainfall estimation relations**
Usage of the $Z−R$ relations that are region, weather system, and precipitation specific can
minimize the weather radars' rainfall estimation uncertainties. In $Z = A\ R^b$ relation, The drop size
can be inferred from the coefficient 'A', and the microphysics from exponent 'b'(Atlas et al.,
1999;Steiner et al., 2004;Atlas and Williams, 2003). The TY and NTY rainfall $Z−R$ relations
derived from the linear regression applied to $10*\log10R$, and $Z$, and are provided in Fig. 10. The
coefficient values of $Z−R$ relations are larger in NTY than the TY for stratiform and convective
precipitations, as well as for total rainfall. This variation is due the presence of significant
number of large size drops in NTY to that of the TY rainfall. Moreover, the obtained TY and
NTY rainy days $Z−R$ relations are found to be varying from the default ($Z=300\ R^{1.4}$) and tropical
$Z−R$ relationships ($Z=250R^{1.2}$), which suggests adopting the weather and region-specific $Z−R$
relations.

**3.5 The rainfall rate relationships with $D_m$ and $N_w$**
The normalized intercept parameter and mass-weighted mean diameter can provide the
RSD features, and these parameters were found to show uniqueness with the rainfall rate (Chen
et al., 2016;Janapati et al., 2020). Scatter plots of $D_m$ and $\log_{10}N_w$ versus $R$ for TY and NTY
rainy days are depicted in Fig. 11. For both TY and NTY rainy days, with an increase in rainfall





rates, the distributions of $D_m$ get narrowed. Similar behavior was reported in previous studies on
tropical cyclone and summer season rainfall (Kumar and Reddy, 2013;Wen et al., 2018;Chang et
al., 2009;Janapati et al., 2020;Chen et al., 2019;Wu et al., 2019). For both TY and NTY rainy
days, fewer variations in $D_m$ values for $R > 25$ mm h$^{-1}$ is due to the reaching of RSD  to
equilibrium condition through raindrop breakup and coalescence, (Hu and Srivastava, 1995), and
an increase in number concentration can lead to a further increase in rainfall rates (Bringi and
Chandrasekar, 2001). The non-linear least-squares fitting equations for $D_m$ versus $R$, and $\log_{10}N_w$
versus $R$ are given in Fig. 11. The $D_m$–$R$ relations depicts that the NTY rainy days have a
relatively higher coefficient value than TY  rainy days, and the coefficient value of $N_w$–$R$
relations is higher in TY than NTY rainy days. This feature confirms that the NTY rainy days
have higher $D_m$ and lower $N_w$ values for given rainfall rates than the TY rainy days.

**3.6 *KE–R and KE–$D_m$*  relations**

The raindrops falling from the cloud base reach the ground with a certain amount of

kinetic energy (*KE*), and they can erode the soil from the ground surface. Hence, the raindrops
*KE* or rainfall *KE* is one of the critical physical quantities in soil erosion studies (Wischmeier,
1959;Kinnell, 1981). As the rainfall *KE* is related to the raindrop diameter and its fall velocity, it
can be evaluated through the RSD information (Kinnell, 1981). The empirical relations between
the rainfall *KE* and rainfall intensity are incorporated in assessing the rainfall erosivity factor (R-
factor) that is used in soil erosion modeling studies (Renard et al., 1997;Janapati et al., 2019). In
this section, empirical relations between the rainfall *KE* (*$KE_{time}$* in J m$^{-2}$ h$^{-1}$; *$KE_{mm}$* in J m$^{-2}$
mm$^{-1}$) and rainfall rate (mm h$^{-1}$) are derived using non-linear least-squares regression method
for both TY and NTY rainy days. The distribution plots of *$KE_{mm}$* and *$KE_{time}$* with $R$ for TY and





NTY rainy days are portrayed in Fig. 12. The $KE_{time}$–$R$ empirical relations are derived by fitting
the data points with power and liner methods. For both TY and NTY rainy days, the power-law
line fitted well by passing through the middle of the data points at both lower and higher rainfall
rates than the linear fit line (Fig. 12a & b).  The $KE_{mm}$ and $R$ data points are fitted with power,
logarithmic, and exponential law. Among three forms of relations, the power-law fitted well with
the data points for both TY and NTY rainy days (Fig. 12c &d). Moreover, empirical relations
between $D_m$ (mm), the $KE_{mm}$ are evaluated for both TY and NTY rainy and are given in Fig. 13.
Comparison of present $KE$−$D_m$ relations with the East China seasonal rainfall $KE$−$D_m$ ($KE =$
$-2.33D_m^2 +21.05D_m -7.79$) relation showed that both TY and NTY relations in Taiwan are
different from that of East China (Wen et al., 2019). The derived $KE$−$D_m$ relations can be used to
estimate the $KE$ values from the remote-sensing radar (GPM DPR) measurements. The $KE_{time}$–$R$,
$KE_{mm}$–$R$, and  $KE$−$D_m$ relations and their statistical values are given in Table 2. For both $KE_{time}$–
$R$, $KE_{mm}$–$R$ relations, the power-law showed higher CC and lower RMSE and NRMSE values,
suggesting to adopt the power form equation to estimate the rainfall $KE$.

**4. Discussion**

To understand the possible mechanisms responsible for the RSD distinctions between TY

and NTY rainy days, re-analysis, remote sensing, and ground-based radar data sets are used. The
water vapor and CAPE values for TY and NTY rainy days over the disdrometer measurement
site are depicted with a box plot in Fig. 14. The water vapor and CAPE values are more
significant in NTY than TY rainy days, suggesting that NTY rainfall events have severe
convective activity with vigorous updrafts and downdrafts than TY rainy days. However, if we
look at the storm and bright band heights (BBH) (Fig. 15), TY rainfall events have relatively





higher BBH than NTY rainy days, and there is no much difference in storm heights between TY
and NTY rainy days. The ice and liquid particles cloud effective radii are illustrated in Fig. 16.
Relatively higher BBH in TY supports the higher CER values of ice particles in TY than NTY
rainy days (Fig. 16b). On the other hand, there is no much difference in the CER values of the
liquid particle between TY and NTY rainy days (Fig.16a). The deep stratiform clouds in TY
rainy days offer sufficient time to the ice crystals to grow larger size (via aggregation and vapor
deposition). They yield raindrops of large size after the passage of the melting layer. Because of
the relatively higher BBH in TY rainfall events, below to the melting layer, the equilibrium RSD
will be achieved through the different microphysical processes (collision, coalescence, and
breakup) in TY than NTY rainfall days (Hu and Srivastava, 1995). In contrast, higher convective
activity in NTY rainy days proposes that the RSD in these intense clouds varies by enhancing the
collision-coalescence process and drop sorting. It allows the smaller drops to shoot to higher
altitudes through resilient updraft and allowing large drops to reach the surface. The relative
humidity and air temperature profiles are portrayed in Fig. 17. The profile of air temperature
shows no much variation between TY and NTY rainy days (Fig.17a). On the other hand, relative
humidity vertical profile values are higher in TY than NTY rainfall, suggesting that the NTY
rainy days were associated with relatively drier conditions than TY days. The rate of evaporation
of small drops produced through the collision breakup processes in NTY is higher than TY rainy
days resulting in more large drops in NTY rainy days.

The radar reflectivity CFAD (contoured frequency-by-altitude diagrams) for (a) typhoon

(TY) and (b) non-typhoon (NTY) rainy days of summer seasons are given in Fig. 18. The
horizontal dotted lines in Fig.18 are the freezing level heights that are computed from the





radiosonde data from Banqiao (121.441°E, 24.997°N) and Hualien (121.619°E, 23.989°N)
stations. The horizontal sky blue (dark magenta) dotted line in Fig.18a (Fig.18b) is melting layer
height mean of TY (NTY) rainy days, and the white dotted line is the mean of both TY and NTY
melting layer heights. The vertical sky blue (dark magenta) star line in Fig. 18a (Fig.18b) is the
mean radar reflectivity profile of TY (NTY) rainy days. The white-star dotted profile in Fig.18a
& b is the mean of both TY and NTY rainy days' reflectivity profiles. The mean reflectivity
profile of TY (NTY) rainy days is less (higher) than the mean of TY and NTY rainy days'
reflectivity profile. A higher occurrence percentage of lower $Z$ values ($Z < 10$ dBZ) in TY than
NTY rainy days can be seen at higher altitudes. In contrast to that, below the melting layer, the
occurrence percentage of higher reflectivity values is ($Z > 40$ dBZ) is higher in NTY than TY
rainy days. The mean vertical profiles of radar reflectivity for TY and NTY rainy days are
plotted in Fig. 19. It can be seen that the mean reflectivity values are higher in NTY than TY
rainy days. As the radar reflectivity is directly related to the raindrop diameter of power six, from
the reflectivity profiles, it can be inferred that bigger drops are predominant in NTY than TY
rainy days. Those mentioned above thermodynamical and microphysical processes resulted in
more big size drops and few small drops in NTY than TY rainy days, resulting in higher $D_m$ and
lower $N_w$ values in NTY than TY rainy days.

**5.  Summary and conclusions**

The summer seasons typhoon (TY) and non-typhoon (NTY) rainy days raindrop size

distributions (RSD) are investigated using long-term (2004-2016) ground-based disdrometer
measurements that were measured in north Taiwan. In addition to disdrometer, remote-sensing,
re-analysis, and ground-based radar data sets are used to elucidate the feasible mechanism



responsible for RSD distinctions in TY and NTY rainfall. The NTY rainy days of summer
seasons have more big size drops and fewer small size drops than the TY rainy days. The
classification of RSD to varying rainfall rates and precipitation regimes clearly showed larger $D_m$
values and smaller $N_w$ values in NTY than TY rainy days. The $Z-R$, $D_m-R$, $N_w-R$, $KE_{time}-R$,
$KE_{mm}-R$, and $KE_{mm}-D_m$ relations were different for TY and NTY rainfall. Relatively higher
convective activity with drier conditions in NTY than TY rainy days resulted in distinct RSD
features between these two weather systems.  The results detailed in this study could help in
evaluating the radar precipitation estimation algorithms, cloud modeling, and rainfall erosivity
studies.

*Data availability.* The Era-interim re-analysis data can be obtained from
https://www.ecmwf.int/en/forecasts/datasets/reanalysis-datasets/era-interim. The TRMM data
can be retrieved from https://gpm.nasa.gov/data/directory. The MODIS cloud data product can
be accessed through https://modis.gsfc.nasa.gov/data/dataprod/mod06.php. The ground-based
radar and disdrometer data are available from the corresponding author upon reasonable request.

*Author contributions.* JJ and BKS conceptualized the idea; PLL and EJ provided funding
acquisition, project administration, and observation data; JJ, BKS, and MTL conducted the
detailed analysis; PLL, and EJ supervised the analysis; JJ, BKS wrote the initial manuscript; JJ,
BKS, PLL reviewed and revised the manuscript; all authors involved in writing the manuscript
and revisions.

*Competing interests.* No conflict of interest is declared by the all authors.




*Acknowledgements.* We acknowledge the Central Weather Bureau (CWB) of Taiwan, in
facilitating the radar reflectivity data, and Tropical Rainfall Measuring Mission (TRMM), ERA-
Interim and MODIS research team for their efforts in providing the data. This research work is
carried out under the Taiwan Ministry of Science and Technology (MOST) grant numbers:
MOST 108-2111-M-008-028, MOST 108-2625-M-008-011, MOST: 104-2923-M-008-003 and
partially by "Earthquake-Disaster & Risk Evaluation and Management Center, E-DREaM" from
The Featured Areas Research Center Program within the framework of the Higher Education
Sprout Project by the Ministry of Education (MOE), Taiwan. The first author, JJ is supported by
the grant number MOST 108–2811–M–008–558, and second author, BKS, by MOST 108-2625-
M-008-011 and MOST 108-2811-M-008-595.



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





**Table 1.** Rainy minutes (N), mean, standard deviation (Std), Skewness and Kurtosis of seven
rainfall rate classes for typhoon (TY) and non-typhoon (NTY) rainy days of summer
seasons.

| Rain rate class | Rain rate threshold | Typhoon (TY) | | | | | non-typhoon (NTY) | | | | |
|---|---|---|---|---|---|---|---|---|---|---|---|
| | | No. of samples | Mean ( mm h$^{-1}$) | Standard deviation (mm h$^{-1}$) | Skewness | Kurtosis | No. of samples | Mean ( mm h$^{-1}$) | Standard deviation (mm h$^{-1}$) | Skewness | Kurtosis |
| C1 | $0.1 \leq R < 1$ | 9317 | 0.43 | 0.26 | 0.55 | 2.1 | 10661 | 0.4 | 0.25 | 0.71 | 2.34 |
| C2 | $1 \leq R < 2$ | 3274 | 1.44 | 0.29 | 0.24 | 1.84 | 3193 | 1.43 | 0.29 | 0.29 | 1.88 |
| C3 | $2 \leq R < 5$ | 4747 | 3.29 | 0.85 | 0.31 | 1.92 | 3404 | 3.17 | 0.83 | 0.46 | 2.1 |
| C4 | $5 \leq R < 10$ | 2799 | 7 | 1.4 | 0.43 | 2.04 | 1404 | 6.98 | 1.42 | 0.43 | 2.01 |
| C5 | $10 \leq R < 30$ | 2313 | 16.44 | 5.24 | 0.77 | 2.59 | 1234 | 17.46 | 5.6 | 0.5 | 2.08 |
| C6 | $30 \leq R < 50$ | 393 | 38.31 | 5.73 | 0.37 | 1.92 | 320 | 37.88 | 5.67 | 0.45 | 2.01 |
| C7 | $R > 50$ | 231 | 67.15 | 14.91 | 1.16 | 3.97 | 152 | 65.86 | 14.94 | 1.51 | 5.18 |
| total | | 23074 | 4.88 | 9.38 | 4.59 | 31.51 | 20368 | 3.59 | 8.38 | 5.2 | 38.9 |













**Table 2.** Statistical parameters [correlation coefficient: $R^2$, Root mean square error (RMSE),
normalized RMSE] for typhoon (TY) and non-typhoon (NTY) rainy days.

| Weather system | Statistical parameter | $KE_{time}-R$ | | $KE_{mm}-R$ | | | $KE_{mm}-D_m$ |
|---|---|---|---|---|---|---|---|
| | | Linear | Power | Power | Exp | Log | Second order polynomial |
| TY | $R^2$ | 0.986 | 0.994 | 0.694 | 0.68 | 0.68 | 0.992 |
| | RMSE | 37.488 | 24.785 | 3.973 | 10.227 | 4.047 | 12.396 |
| | NRMSE | 0.306 | 0.202 | 0.032 | 0.083 | 0.033 | 2.514 |
| NTY | $R^2$ | 0.984 | 0.99 | 0.646 | 0.639 | 0.639 | 0.988 |
| | RMSE | 38.012 | 30.745 | 4.599 | 11.017 | 4.636 | 12.93 |
| | NRMSE | 0.322 | 0.26 | 0.039 | 0.093 | 0.039 | 2.803 |



















**Figures**

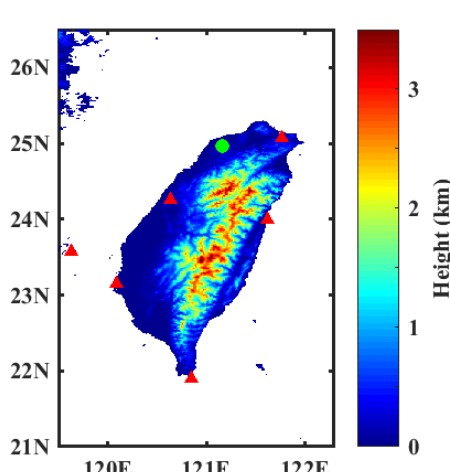


**Figure 1.** Taiwan geography with the location of disdrometer (green color circle) and radars

(red color triangles).


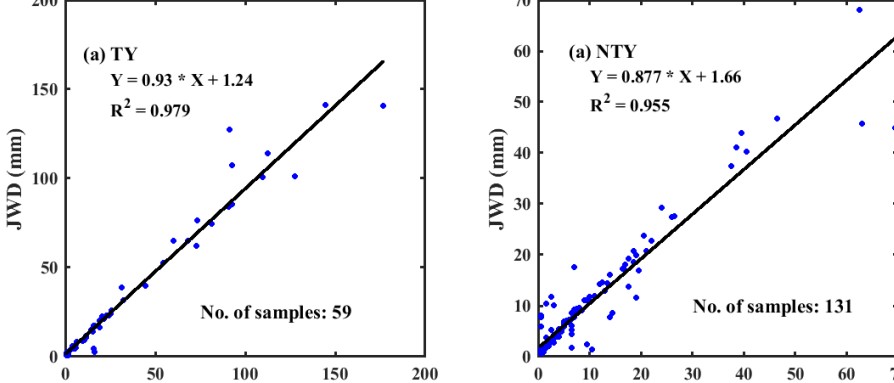


**Figure 2**. Comparison JWD measured daily accumulated rainfall amounts with the collocated

rain gauge for (a) typhoon (TY) and (b) non-typhoon (NTY) rainy days of summer

seasons.







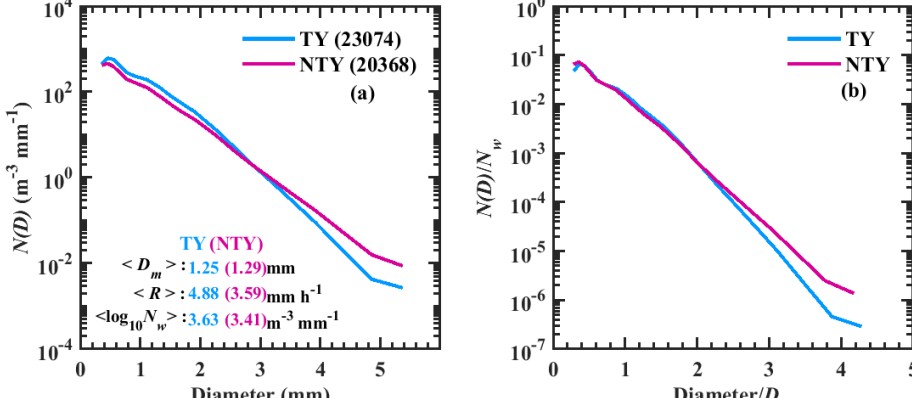


**Figure 3**. (a) Distributions of mean concentration [$N(D)$, in mm$^{-1}$ m$^{-3}$] with raindrop diameter

for typhoon (TY) and non-typhoon (NTY) rainfall and their (b) normalized spectra.




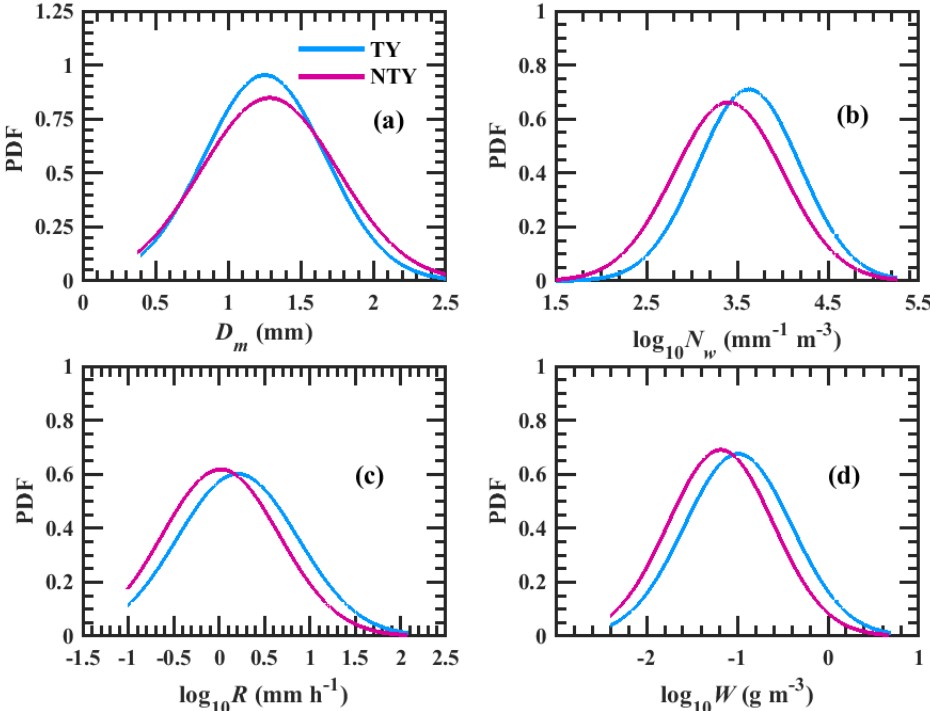


**Figure 4.** (a) mass-weighted mean diameter, $D_m$ (mm), (b) $\log_{10}N_w$, where $N_w$ is the normalized

intercept parameter (mm$^{-1}$ m$^{-3}$) (c) $\log_{10}R$, where $R$ is rainfall rate (mm h$^{-1}$) (d) $\log_{10}W$,

where $W$ is liquid water content (g m$^{-3}$) probability distribution functions (PDF) for

typhoon (TY) and non-typhoon (NTY) rainy days of summer seasons.

681



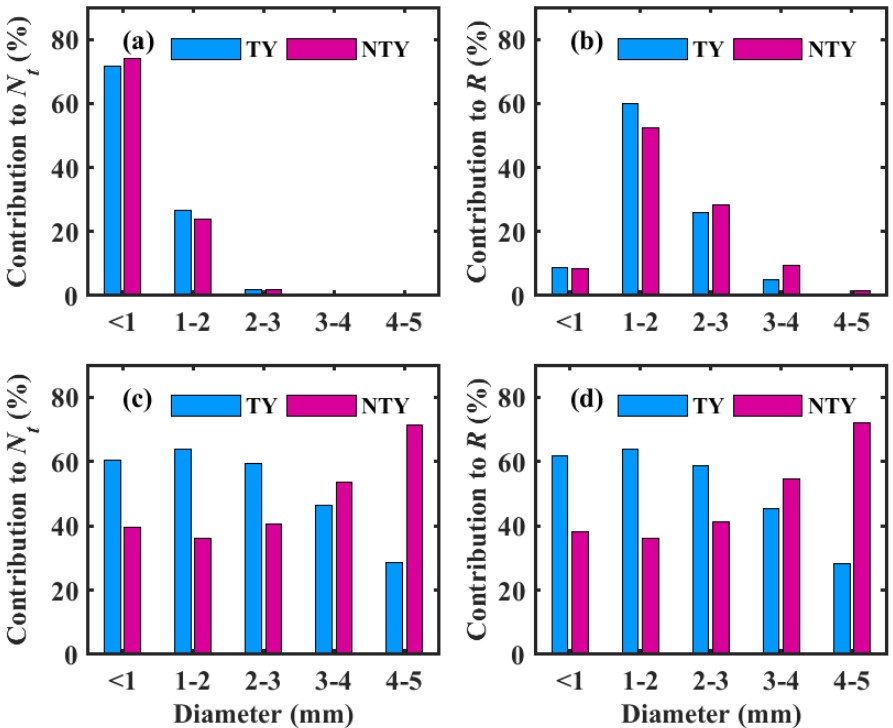

682

**Figure 5.** Contribution of drop diameter classes (Diameter < 1 mm, 1−2 mm, 2−3 mm, 3−4 mm, and 4−5 mm) to (a) total number concentration $N_t$ (m$^{-3}$) and (b) rainfall rate $R$ (mm h$^{-1}$) in typhoon (TY) and non-typhoon (NTY) rainfall of summer seasons. Percentage of typhoon (TY) and non-typhoon (NTY) rainfall in each diameter class for (c) total number concentration $N_t$ (m$^{-3}$) and (d) rainfall rate $R$ (mm h$^{-1}$).

688





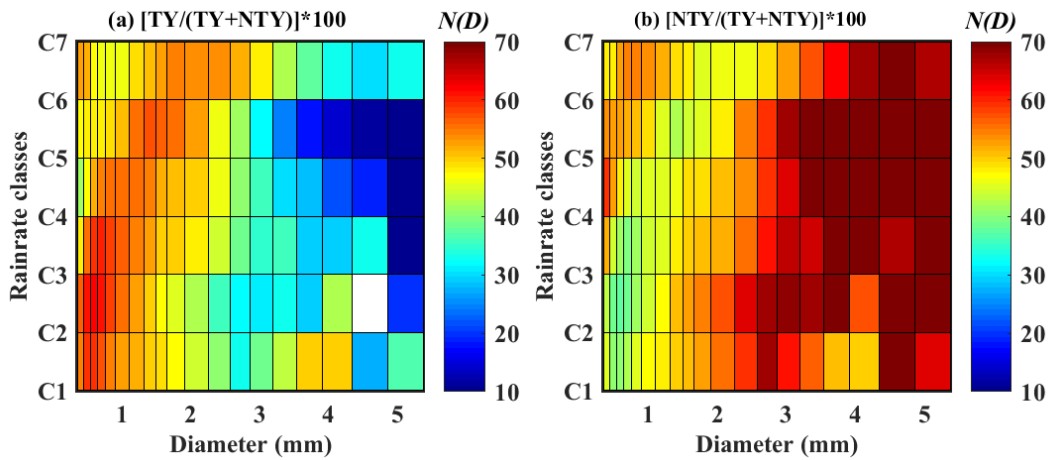

689

**Figure 6** Percentage contribution of raindrop concentration [$N(D)$, mm$^{-1}$ m$^{-3}$] in different

rainfall rate ranges for typhoon (TY) and non-typhoon (NTY) rainfall of summer seasons.



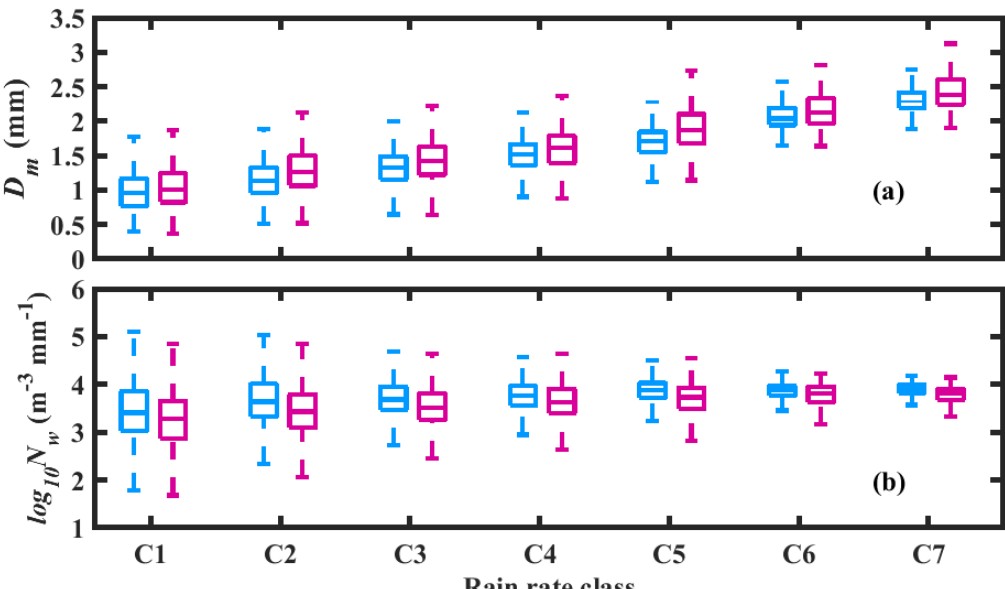

**Figure 7.** Box plot of (a) $D_m$ (mm) and (b) $\log_{10}N_w$ (mm$^{-1}$ m$^{-3}$) in seven rainfall rate intervals for typhoon (TY) (sky blue color) and non-typhoon (NTY) (dark magenta color) rainfall. The center line of the box indicates the median, and the bottom and top lines of the box indicate the 25$^{th}$ and 75$^{th}$ percentiles, respectively. The bottom and top of the dashed vertical lines indicate the 5$^{th}$ and 95$^{th}$ percentiles, respectively.

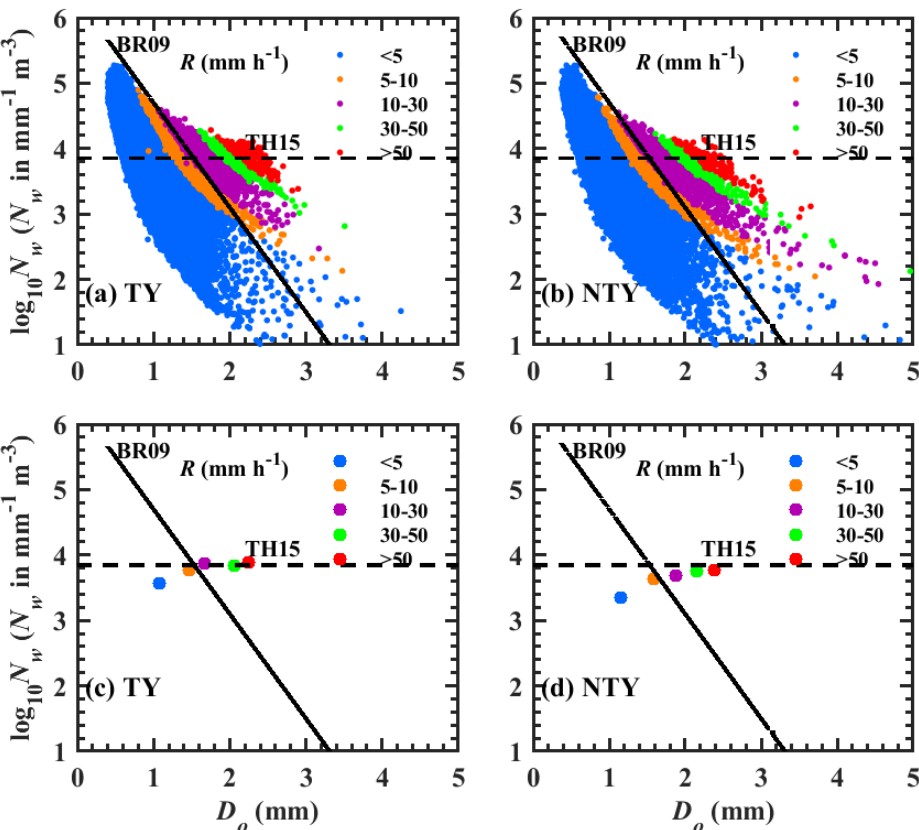

703

**Figure 8.** Scatter plots of $D_0$-$\log_{10}N_w$ for (a) typhoon (TY) and (b) non-typhoon (NTY) rainfall in

different rainfall rate ranges. Mean values of $D_0$ and $\log_{10}N_w$ for (c) typhoon (TY) and (d)

non-typhoon (NTY) rainfall in different rainfall rate ranges. Stratiform and convective

regimes separation line of Thompson et al. (2015): TH15 and Bringi et al. (2009): BR09

are represented with horizontal dotted line and inclined solid line, respectively.

709

710





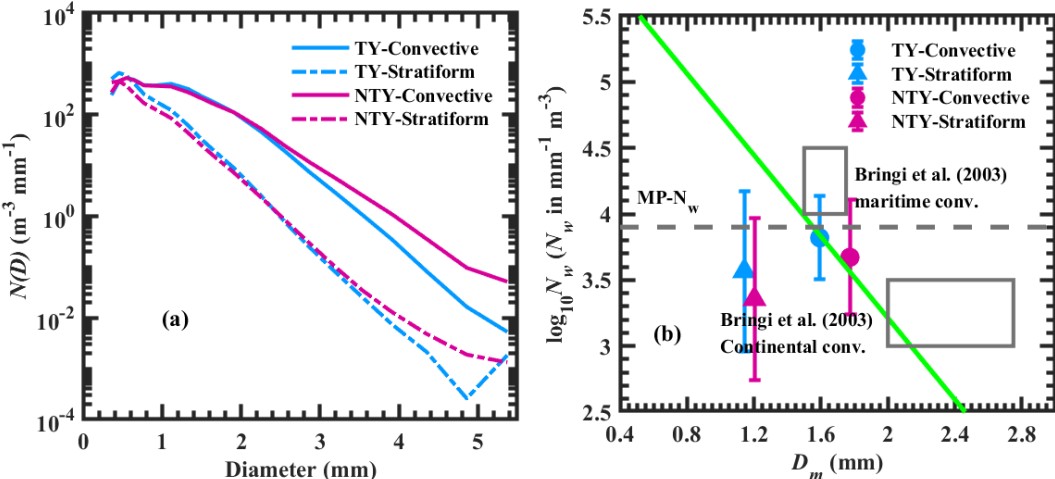

**Figure 9**. (a) Distribution of $N(D)$ ($m^{-3}$ $mm^{-1}$) with raindrop diameter in stratiform and convective regimes for typhoon (TY) and non-typhoon (NTY) rainfall. (b) Variations of $\log_{10}N_w$ (where $N_w$ is the normalized intercept parameter in $mm^{-1}$ $m^{-3}$) with $D_m$ (mass-weighted mean diameter in mm) values in stratiform and convective regimes for typhoon (TY) and non-typhoon (NTY) rainfall. The horizontal gray dashed line is the Marshall-Palmer value of $\log_{10}N_w$ (3.9) for exponential shape. The green dash dotted line is the stratiform and convective separation line of Bring et al. (2003).

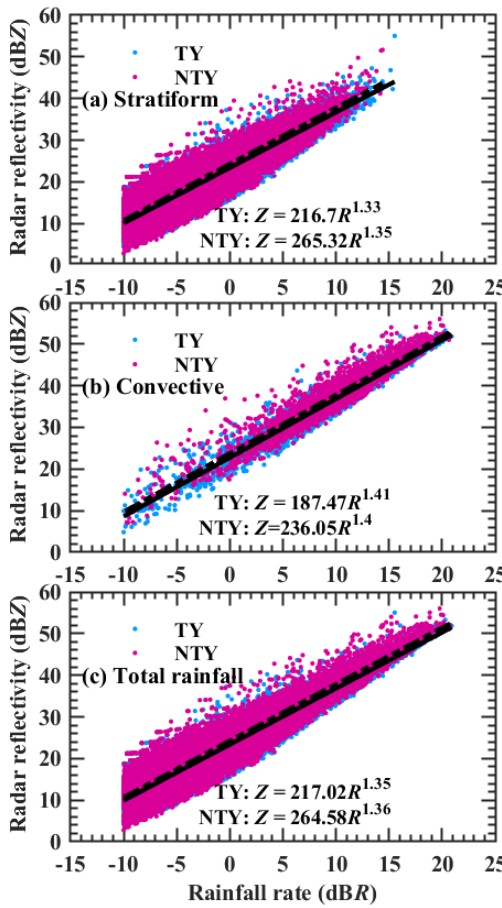


**Figure 10.** Scatter plots of radar reflectivity ($Z$, dB$Z$) and logarithmic scale of rainfall rate (10*$\log_{10}R$, dB$R$, $R$ in mm h$^{-1}$) for typhoon (TY) and non-typhoon (NTY) rainfall.





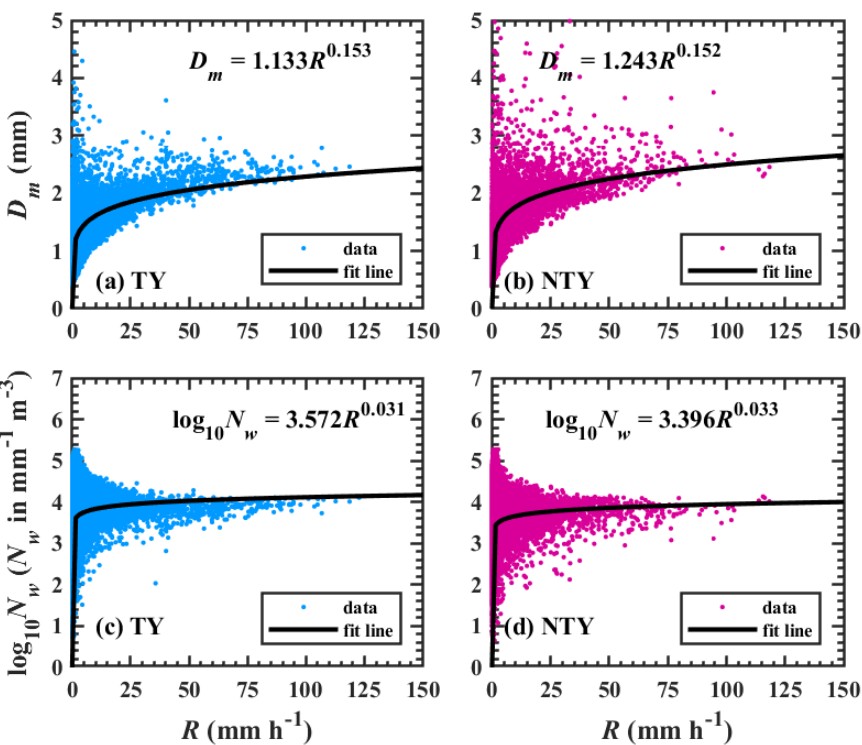


**Figure 11.** Distributions of $D_m$ (mm) and $\log_{10}N_w$ $N_w$ in mm$^{-1}$ m$^{-3}$) with rainfall rate ($R$, mm h$^{-1}$)

for typhoon (TY) and non-typhoon (NTY) rainy days of summer seasons.

728 .









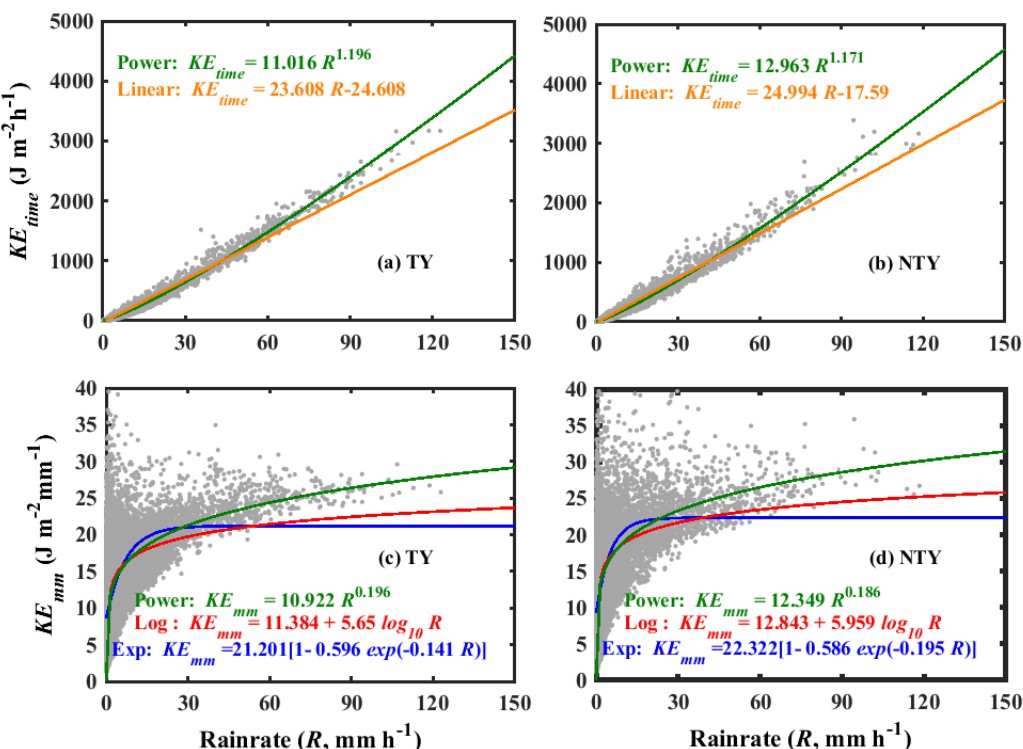

**Figure 12.** Scatter plots of rainfall kinetic energy (*KE*) [time-specific *KE*, $KE_{time}$; volume-specific *KE*, $KE_{mm}$] with rainfall rate (*R*, mm h$^{-1}$) for typhoon (TY) and non-typhoon (NTY) rainy days of summer seasons.






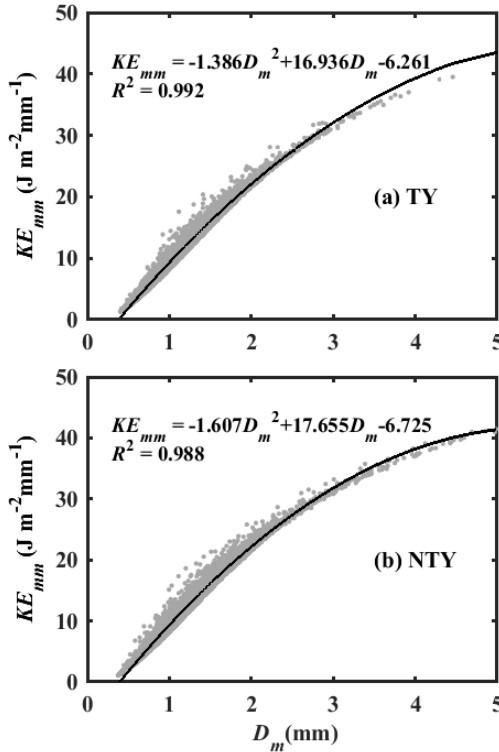


**Figure 13.** Scatter plots of volume-specific *KE (KE$_{mm}$* in J m$^{-2}$ mm$^{-1}$] with *D$_m$* (mm) for typhoon

(TY) and non-typhoon (NTY) rainy days of summer seasons.






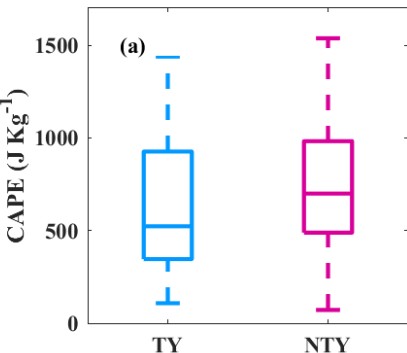
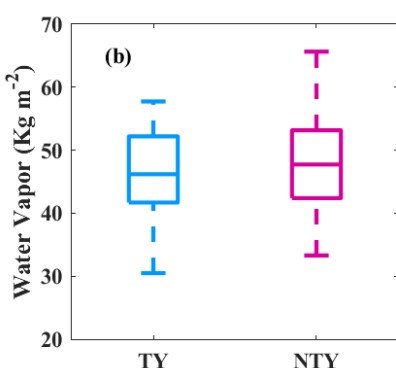

**Figure 14:** Variations in (a) convective available potential energy (CAPE, J Kg$^{-1}$) and (b) vertical integral of water vapor (kg m$^{-2}$) for typhoon (TY) and non-typhoon (NTY) rainy days of summer seasons over disdrometer observational site. The center line of the box indicates the median, and the bottom and top lines of the box indicate the 25$^{th}$ and 75$^{th}$ percentiles, respectively. The bottom and top of the dashed vertical lines indicate the 5$^{th}$ and 95$^{th}$ percentiles, respectively

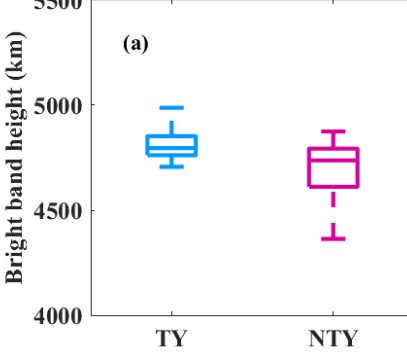
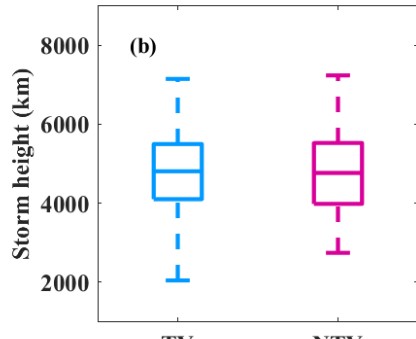

**Figure 15.** (a) bright band (BB) and storm heights box plots for typhoon (TY) and non-typhoon (NTY) rainy days of summer seasons over disdrometer observational site.







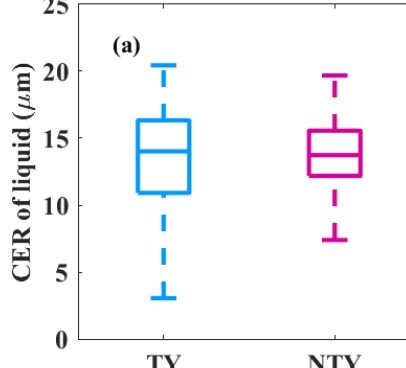 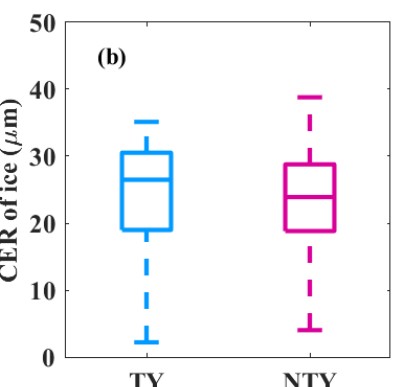


**Figure 16.** (a) liquid, (b) ice particles cloud effective radii (CER, μm) values for typhoon (TY)

and non-typhoon (NTY) rainy days of summer seasons over disdrometer observational

site.







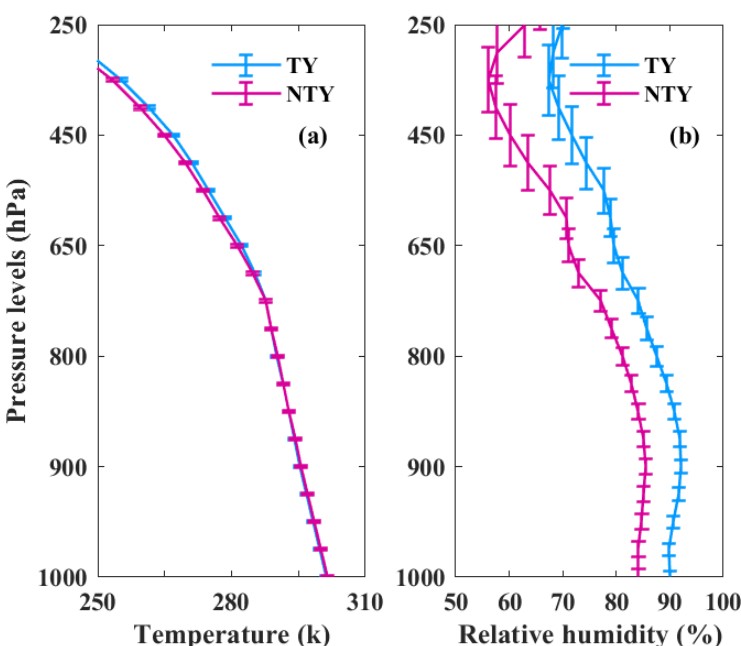

**Figure 17.** Mean air temperature (°C) and relative humidity (%) profiles for typhoon (TY) and non-typhoon (NTY) rainy days of summer seasons over the disdrometer observational site.





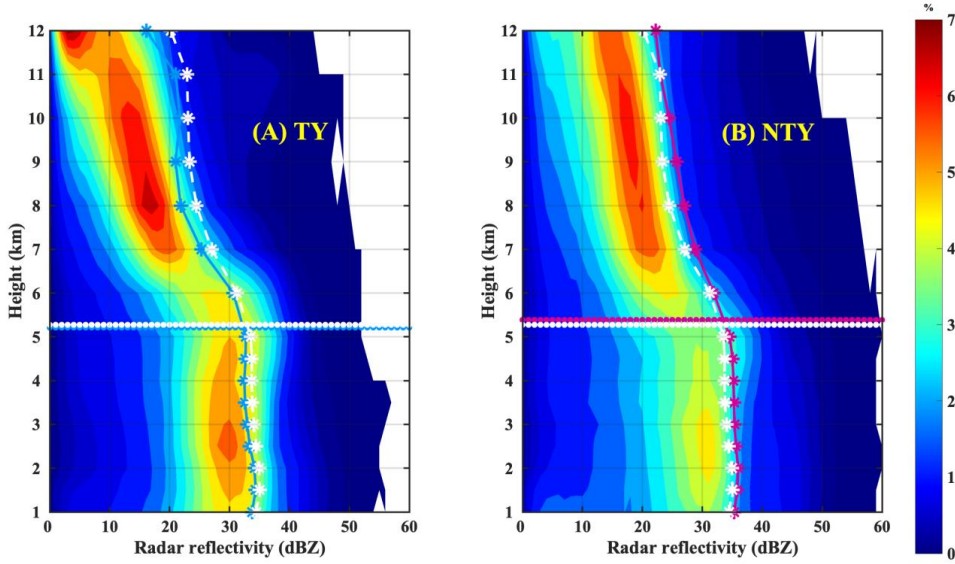


**Figure 18.** Contoured frequency-by-altitude diagram of radar reflectivity from six ground-based
radars for (a) typhoon (TY) and (b) non-typhoon (NTY) rainy days of summer seasons.





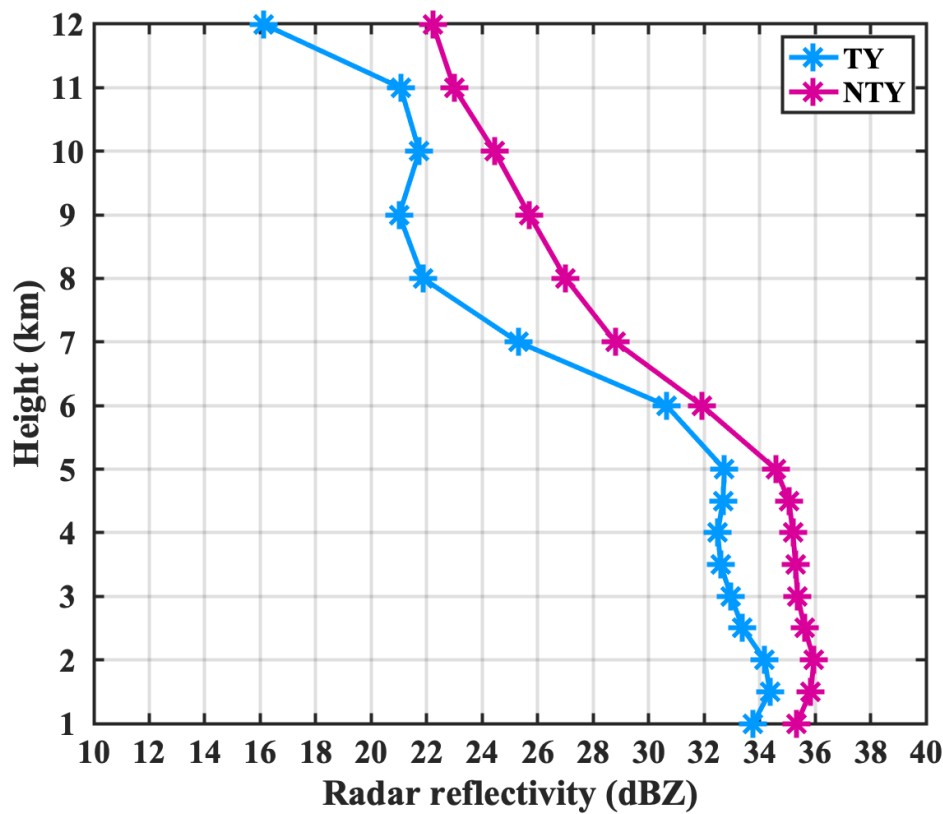

**Figure 19.** Mean radar reflectivity profiles of typhoon (TY) and non-typhoon (NTY) rainy days

of summer seasons over the disdrometer observational site.