# Peer review of "Microphysical features of typhoon and non-typhoon rainfall observed in Taiwan, an island in the northwest Pacific."

_Hydrology and Earth System Sciences, 2020_

## Referee Comment (RC1) · Anonymous Referee #1 · 16 Sep 2020

I am sorry to post a very negative review for this paper. The reason is that the work and results are very interesting and try to fill a gap in observation of RSD in a tropical environment. However, I tried to read the paper, but the quality of the language is too low to ensure a thorough review. It is not the duty of a reviewer to polish the text for grammar flaws. Without polishing the text is unreadable, at least in my opinion. The rejection is meant to have the authors completely reconsider the text and polish the grammar and style. I strongly encourage a resubmission after language polishing.

---

## Referee Comment (RC2) · Anonymous Referee #2 · 21 Sep 2020

The paper "Microphysical features of typhoon and non-typhoon rainfall observed in Taiwan, an island in the northwest Pacific", by Janapati and co-workers, presents a study based on disdrometric data, aiming to describe precipitation characteristics in case of rain produced by Typhoons over Taiwan.

A large Joss-Waldvogel (2009-2017) disdrometer dataset is separated in Typhoon and no-Typhoon samples, that are analysed to highlight similarity and differences between the two subsets, also considering other data such as reanalysis and weather radar data.

The subject is interesting and the Authors did a significant work in processing such a

large amount of data. However, I think that the manuscript should go under a major revision, for a number of reasons that I list below.

First, the writing is extremely poor: in many cases the reader cannot understand the text. I suggest a deep language revision of the manuscript.

Second, the J-W disdrometer has a number of known deficiencies (see Tokay A, Kruger A, Krajewski WF. Comparison of drop size distribution measurements by impact and optical disdrometers. J Appl Meteor 2001;40:2083–97 among many others), especially in case of heavy rain, that should be reported and discussed in detail.

The analysis of CAPE, water vapour and temperature profiles seems a bit out of context here. The paper deals with precipitation microphysical structure, and these environmental quantities are not so relevant to the whole analysis. I suggest to drop this part of the work.

The conclusions are very weak and should be more deep, reporting main results, and not simply saying "...relations were different for TY and NTY rainfall". There is a recent paper by Bao and co-workers (Distinct Raindrop Size Distributions of Convective Inner‐and Outer‐Rainband Rain in Typhoon Maria (2018), Journal of Geophysical Research: Atmospheres, 125, e2020JD032482. https://doi.org/10.1029/2020JD032482) that can be useful to comment some result.

Please, put the right units for all the entries in the tables.

---

## Author Comment (AC1) · 9 Nov 2020

We thank the reviewer for providing encouraging and positive opinion on our research results. Proper care has been taken in modifying the complete manuscript in terms of grammar and polishing the text. We hope our revised version of the manuscript is now suitable for the review process.

---

## Author Comment (AC2) · 9 Nov 2020

**Response to referee #2**

**Overall comment:**
*The paper "Microphysical features of typhoon and non-typhoon rainfall observed in Taiwan, an island in the northwest Pacific", by Janapati and co-workers, presents a study based on disdrometric data, aiming to describe precipitation characteristics in case of rain produced by Typhoons over Taiwan. A large Joss-Waldvogel (2009-2017) disdrometer dataset is separated in Typhoon and no-Typhoon samples, that are analysed to highlight similarity and differences between the two subsets, also considering other data such as reanalysis and weather radar data. The subject is interesting and the Authors did a significant work in processing such a large amount of data. However, I think that the manuscript should go under a major revision, for a number of reasons that I list below.*

**Response:** We are grateful to the reviewer for providing constructive and positive comments on our manuscript. We tried to modify our manuscript as per the reviewer's suggestions, and the responses to individual comments are given below.

**Specific comments:**

1. *First, the writing is extremely poor: in many cases the reader cannot understand the text. I suggest a deep language revision of the manuscript.*

**Response:** We thank the reviewer for this comment. We have thoroughly modified the complete manuscript to minimize the grammatical errors, and the revised version of the manuscript is submitted.

2. *Second, the J-W disdrometer has a number of known deficiencies (see Tokay A, Kruger A, Krajewski WF. Comparison of drop size distribution measurements by impact and optical disdrometers. J Appl Meteor 2001;40:2083–97 among many others), especially in case of heavy rain, that should be reported and discussed in detail.*

**Response:** We are thankful to the reviewer for providing this comment. We have done some quality control of the JWD data before using it for further analysis. However, we didn't mention it in the initial version. A brief description of the JWD deficiencies and the JWD data quality control is provided (as given below) in the revised manuscript (second paragraph of section 2: lines 122-138).

"The JWD has its advantage and disadvantages over the other disdrometers (Lee and 122 Zawadzki, 2005;McFarquhar and List, 1993;Sauvageot and Lacaux, 1995;Sheppard, 1990;Sheppard and Joe, 1994;Tokay et al., 2001;Tokay et al., 2013). For instance, JWD can't measure fall velocity; hence, to evaluate the RSD parameters from the JWD, we assumed that raindrops reach the ground with terminal velocity. Further, in heavy rainfall events, the JWD measures the spurious values for the raindrops of diameter < 1 mm, and it was named as the

dead-time of the instrument. To deal with the dead-time of the JWD, the manufacturer provided an error correction multiplication matrix based on a correction scheme from Sheppard and Joe (1994). However, as the JWD can't record any drops for the first three to four channels in heavy rainfall events, the multiplicative matrix algorithm does not increase the counts when the channel has no drops (Tokay & Short, 1996; Tokay et al., 2001); hence, in this study, we didn't apply the dead-time correction to the JWD data. On top of that, 1-min RSD samples with raindrops count < 10 and rainfall rate < 0.1 mm h-1 were discarded (Tokay & Short, 1996). The daily rainfall accumulations from the JWD are related to the collocated rain gauge for both TY and NTY rain regimes and are illustrated with scatter plots in Fig.2. Strong correlations between JWD and rain gauge measurements for both TY and NTY days provide the trustworthiness of the JWD data for further analysis".

3. *The analysis of CAPE, water vapour and temperature profiles seems a bit out of context here. The paper deals with precipitation microphysical structure and these environmental quantities are not so relevant to the whole analysis. I suggest to drop this part of the work.*

**Response:** We thank the reviewer for this comment. In the initial version of the manuscript (line nos.127-129) we mentioned

 "Along with the disdrometer data, remote-sensing (TRMM and MODIS) and reanalysis (ERA-interim) data sets are used to elucidate the typhoon and non-typhoon rainy days' microphysical characteristics", and this typographic error is rectified by modified this sentence as

"In addition to disdrometer data, remote-sensing (TRMM and MODIS) and reanalysis (ERA-interim) data sets are used to elucidate the thermodynamical and microphysical characteristics that are accountable for the possible disparities in RSDs between TY and NTY rainfall." at line no. 149-152 of the revised manuscript

As we tried to report the thermodynamical and microphysical processes that are responsible for the RSDs variations between typhoon and non-typhoon rainy days, we continue to keep this part of the work in the revised version too. We once again thank the reviewer in understanding our reasoning for not dropping this section in the revised manuscript.

*The conclusions are very weak and should be more deep, reporting main results, and not simply saying ": : :relations were different for TY and NTY rainfall". There is a recent paper by Bao and co-workers (Distinct Raindrop Size Distributions of Convective Innerˇ A ˘ Rand OuterˇARˇ Rainband Rain in Typhoon Maria (2018), Journal of Geophysical Research: Atmospheres, 125, e2020JD032482. https://doi.org/10.1029/2020JD032482) that can be useful to comment some result. Please, put the right units for all the entries in the tables.*

**Response:** We are grateful to the review for providing this comment. We tried to enhance the conclusion section by providing more details in the summary and conclusion section of the revised manuscript (as given below at line no. 391-415).

"Raindrop size distributions (RSDs) of typhoon (TY) and non-typhoon (NTY) days have been analyzed using long-term (2004-2016) disdrometer measurements from north Taiwan. Along with disdrometer data, other auxiliary (remote-sensing, re-analysis, and ground-based radar) data sets have been used to elucidate the feasible mechanisms liable for the distinctions in RSDs concerning TY and NTY rainfall. The NTY days have more big size drops and less small size drops than TY days, resulting in larger $D_m$ and smaller $N_w$ values in NTY days. Likelihood for the diverse microphysical processes between TY and NTY rainfall is exemplified by exclusive separation in TY and NTY rainfall normalized raindrop spectra at $D/D_m > 2$. The classification of RSDs to varying rainfall rates and precipitation (stratiform and convective) regimes clearly show smaller $D_m$ and larger $N_w$ values in TY than NTY days. The percentage contribution of large (small and mid-size) drops to $N_t$ and $R$ is lower (higher) in TY than NTY rainfall. For both TY and NTY rainy days, stratiform precipitations $D_m$ and $N_w$ values are smaller than the maritime and continental clusters, while, convective precipitations $D_m$ values are approximately within the range of maritime clusters. The rainfall kinetic energy and intensity ($KE_{time}-R$ and $KE_{mm}-R$) relations evaluated for both TY and NTY rainy days reveal greater performance of power relation than other types, and confirms to use power form of $KE-R$ relations in assessing the rainfall erosivity factor for TY and NTY rainfall events. The enumerated $Z-R$, $D_m-R$, $N_w-R$, $KE_{time}-R$, $KE_{mm}-R$, and $KE_{mm}-D_m$ relations showed profound diversity between TY and NTY rainfall and substantiate the significance of adopting precipitation specific empirical relations in evaluating the rainfall rate and kinetic energy values. Overall, present study confirms that relatively higher convective activity with drier conditions in NTY than TY days significantly wedged the disparities in RSDs with dissimilar microphysical processes. The current observational outcomes could benefit in appraising the radar precipitation estimation algorithms, cloud modeling, and rainfall erosivity in north Taiwan for TY and NTY rainfall events."

The discussion on Bao et al. (2020) results is provided in the introduction (section 1 at line no. 46-50) and results (section 3.2: line no. 240-245, and Section 3.4: line no. 299-301) sections of the revised manuscript.

"Owing to the aforementioned implications of RSDs, ample literature exists on RSDs for spatial, seasonal (Thompson et al., 2015;Jayalakshmi and Reddy, 2014;Seela et al., 2017;Seela et al., 2018;Krishna et al., 2016;Seela et al., 2016) variations, storm to storm, within the storm (Kumari et al., 2014;Maki et al., 2001;Jung et al., 2012;Bao et al., 2020;Janapati et al., 2017), and different precipitations (Tokay and Short, 1996;Krishna et al., 2016)."

"As can be seen from Fig. 7a, with the increase in rainfall rate class, $D_m$ values increase for both TY and NTY rainfall, which is due to a raise in large size drops concentration and a reduction in small drops concentration (Rosenfeld and Ulbrich, 2003;Krishna et al., 2016), and similar finding were noticed by previous researchers for both tropical cyclones and non-tropical cyclones rainfall (Bao et al., 2020;Deo and Walsh, 2016;Jayalakshmi and Reddy, 2014;Radhakrishna and Narayana Rao, 2010)."

"The current TY rainfall $Z{-}R$ relations show disparity with the other locations tropical cyclones rainfall relations (Bao et al., 2020;Wen et al., 2018;Janapati et al., 2020)."

Missed units for second column parameter (rain rate threshold) of Table 1 are provided in the revised manuscript as mm h$^{-1}$.

---

## Referee Report (RR1)

**REVIEW REPORT**

Review of  hess-2020-345-manuscript-version3

By  Jayalakshmi Janapati, Balaji Kumar Seela, Pay-Liam Lin, Meng-Tze Lee , Everette Joseph

Manuscript Title – Microphysical features of typhoon and non-typhoon rainfall observed in Taiwan, an island in the northwest Pacific

**GENERAL COMMENTS**

The manuscript mainly analyzed RSD data collected in north Taiwan during Typhoon and non typhoon events. Furthermore, a brief analysis of additional data (reanalysis, remote-sensing, and ground-based datasets) is also provided. The manuscript is well written and easy to follow. I suggest the publication on Hydrology and Earth System Sciences after addressing my minor comments.

**SPECIFIC COMMENTS:**

- Lines 192-194: the sentence need to be further explained. Please add some details
- Figure 5: check the y-label of Figure 5c & d
- Lines 221-224: Please explain better the separation criterion for typhoon and non typhoon events, so that it can be easily applied also to other researchers
- Lines 231-232: please write the equation! It will be more clear to the reader
- Line 251: Why the Authors did not use the C1-C7 rain classes in Figure 8 as in Figure 6 and Figure 7?
- Lines 274-276: Why the Authors did not use the Bringi et al. method described and analyzed in the previous section? Please use that method or eliminate it and include Me et al. method in the analysis reported in the previous section of the manuscript or justify why you analyzed your data with respect to the Bringi et al. method and then you used the Ma et al method for classification.
- Line 298: "is due the presence" instead of "is due the presence"
- Line 303: please see Adirosi et al. (2018) for the effects of different disdrometer types on the Z-R relation
- Line 357: please specify the meaning of CER

REFERENCE

Adirosi, E., Roberto, N., Montopoli, M., Gorgucci, E., & Baldini, L. (2018). Influence of disdrometer type on weather radar algorithms from measured DSD: Application to Italian climatology. Atmosphere, 9(9), 360.

---

## Referee Report (RR2)

**REVIEW REPORT**

Review of  hess-2020-345-manuscript-version4

By  Jayalakshmi Janapati, Balaji Kumar Seela, Pay-Liam Lin, Meng-Tze Lee , Everette Joseph

Manuscript Title – Microphysical features of typhoon and non-typhoon rainfall observed in Taiwan, an island in the northwest Pacific

**GENERAL COMMENTS**

Authors addressed all my comments. I have only one minor comment.

**SPECIFIC COMMENTS:**

1) If I understood well, in the answer to my previous comment n6, the one regarding the use of Bringi et al. methods to separate between stratiform and convective rain, the Authors wrote that in Figure 8 of the paper  they use Bringi et al. (2003) classification line as reference and then in the analysis they used the Ma et al. (2019) method that is derived from the Bringi et al. (2003) method. Please note that in the text in section 3.2 and in the caption of Figure 8 the Authors refer to Bringi et al. (2009) and not to Bringi et al. (2003). Furthermore, I understand the need of using Ma et al. (2019) instead of Bringi et al. (2003), but probably, for consistency the line obtained with Ma et al. (2019) can be added to Figure 8, to understand its performance with respect to Bringi et al. (2003).

---

## Author Response (AR2)

**Response to referee #02** (Report #02)

The manuscript has been substantially re-written improving the language and the readability of the text. I'm still not fully convinced about the quality of the language, but, since I'm not a native English speaker, I do not want to focus on this issue. Coming to the scientific side, I see two major issues that the Authors should fix (or, at least, deeply discuss) in the text before the manuscript is accepted for publication on HESS.

**Reply:** We sincerely acknowledge the reviewer for providing positive view and constructive comments on our manuscript. We have addressed all the reviewer's comments by providing detailed response for each comment, and the necessary modifications are also made (with red color text) in the revised manuscript

**Major comments:**

1. First, the distinction between TY and NTY spectra is rather weak (fig. 3), and the Authors should show that the spectra variability within TY and NTY classes is smaller than the differences between averaged TY and NTY spectra. The Authors, for instance, could add to the averaged spectra in Fig. 3 the curves of mean spectra +- one standard deviation. Also other results show very similar values between TY and NTY samples. In the conclusion, the sentence: "Likelihood for the diverse microphysical processes between TY and NTY rainfall is exemplified by exclusive separation in TY and NTY rainfall normalized raindrop spectra at $D/D_m > 2$", is not supported by evidence. There is no "exclusive separation" in figure 3b beyond $D/D_m=2$: we can see that there is an evidence that NTY precipitation has higher occurrence of larger drops.

**Reply:** We thank the reviewer for this comment. As per the reviewer's suggest, Fig.3 is redrawn by providing error bars (± standard deviation) to each drop diameter bin and is given below with Y-axis in logarithmic and linear scale. The figure shows that the spectral variability within TY and NTY classes is smaller than the differences between averaged TY and NTY spectra.

[Figure]

**Figure R1.** Mean raindrop size distributions of TY and NTY rainfall with Y-axis in logarithmic (left panel) and linear (right panel) scale.

We have modified Fig.3a by providing error bars at each diameter bin in the revised manuscript and the below mentioned sentence is incorporated in the revised manuscript at lines 193-196.

"Despite of weak distinction between TY and NTY mean rain spectra for raindrops of diameter < 2 mm, it can be seen that the spectra variability within TY and NTY classes is smaller than the differences between averaged TY and NTY spectra."

As per the reviewer's opinion, the sentence in the conclusion section is modified as "The mean normalized RSD of NTY precipitation has a higher occurrence of larger drops (at $D/D_m > 2$) than TY precipitation, which indicates the possibility for diverse microphysical processes between these two weather conditions." in the revised manuscript at lines 424-427.

2. A second issue is on the JW performances under Typhoon weather, i.e. when rain comes together with strong winds. The hypothesis that drops fall vertically at terminal velocity, does still hold under very strong horizontal winds? There are any measurements of wind speed in TY and NTY days?

**Reply:** We thank the reviewer for this comment. Before considering the JWD measurements in the analysis, we compared daily rainfall amounts from JWD with the collocated rain gauge for both TY and NTY rainy days as shown in Fig.2. The rainy days with large discrepancy between JWD and rain gauge measurements were already discarded in this study, however, we didn't mention this statement in the manuscript. We noticed four TY rainy with larger discrepancy between JWD and rain gauges measurements and were excluded in the analysis, and there were no NTY rainy days with much discrepancy between JWD and rain gauge measurements.

As per the reviewer's suggestion, we compared the daily rainfall amounts of JWD with rain gauge for different wind speed conditions (daily maximum wind speed: 0-8, 8-14, 14-18, > 18 m s$^{-1}$) and the results are given in the below table.

**Table R1.** The JWD and rain gauge comparison results (n: number of rainy days, CC: correlation coefficient, RMSE: root mean square error) for different wind speed conditions (daily maximum wind speed: 0-8, 8-14, 14-18, > 18 m s$^{-1}$). Note: there were no NTY rainy days with daily maximum wind speed > 14 m s$^{-1}$.

| Wind speed | TY | | | NTY | | |
|---|---|---|---|---|---|---|
| (m s$^{-1}$) | n | CC | RMSE (mm) | n | CC | RMSE (mm) |
| 0-8 | 21 | 0.989 | 6.305 | 113 | 0.956 | 3.853 |
| 8-14 | 27 | 0.99 | 5.153 | 18 | 0.942 | 3.482 |
| 14-18 | 8 | 0.953 | 18.112 | - | - | - |
| >18 | 3 | 0.996 | 7.448 | - | - | - |

Below mentioned sentences are added in the revised manuscript at lines 152-159

"The rainy days (TY: 04 days and NTY: 0 days) with larger discrepancy between JWD and rain gauge measurements were discarded in this study. Further, we compared the JWD measurements (for both TY and NTY rainy days) with the rain gauge for different wind speed conditions (daily maximum wind speed: 0-8, 8-14, 14-18, > 18 m s$^{-1}$), and the results are provided in Table 1. For the considered NTY rainy days, the daily maximum wind speeds were less than 14 m s$^{-1}$, however, there were TY rainy days with wind speed > 18 m/s. A good agreement between JWD and rain gauge measurements for both TY and NTY days (Fig.2 and Table 1) provided the trustworthiness of the JWD data for further analysis"

**Table 1.** The JWD and rain gauge comparison results (n: number of rainy days, CC: correlation coefficient, RMSE: root mean square error) for different wind speed conditions (daily maximum wind speed: 0-8, 8-14, 14-18, > 18 m s$^{-1}$). Note: there were no NTY rainy days with daily maximum wind speed > 14 m s$^{-1}$.

| Wind speed (m s$^{-1}$) | TY | | | NTY | | |
|---|---|---|---|---|---|---|
| | n | CC | RMSE (mm) | n | CC | RMSE (mm) |
| 0-8 | 21 | 0.989 | 6.305 | 113 | 0.956 | 3.853 |
| 8-14 | 27 | 0.99 | 5.153 | 18 | 0.942 | 3.482 |
| 14-18 | 8 | 0.953 | 18.112 | - | - | - |
| >18 | 3 | 0.996 | 7.448 | - | - | - |

**Minor comments:**

1. Lines 122-113. "if a typhoon was invaded…" does it mean that the "typhoon center" invaded, or simply some piece of the cloud structure invaded the box?
**Reply:** To provide more clarity to the sentence, we modified "if a typhoon was invaded" with "if a typhoon center was invaded" in the revised manuscript at lines 128-129.

2. Line 175. I do not thing that "formidable" is the right adjective here.
**Reply:** The "formidable" is modified to "difficult" in the revised manuscript at line 197.

3. Line 226. In my opinion "weather system" cannot be an attribute of NTY cases, since many different weather systems can occur in NTY days. I suggest to use "weather condition" or "weather settings", instead.
**Reply:** As per the reviewer's suggestions, we have modified "weather systems" to "weather conditions" throughout the revised manuscript.

4. Lines 393-395. This sentence needs rewriting, in this way is unclear.

**Reply:** As per the reviewer's suggestion, the sentence is modified as "Besides disdrometer data, other auxiliary data sets (remote-sensing, re-analysis, and ground-based radar) have been used to discuss the disparities in RSDs between TY and NTY rainfall." In the revised manuscript at lines 422-423.

5. Table 2. Please, report the unit for RMSE

**Reply:** The units for RMSE are J $m^{-2}$ $h^{-1}$ for $KE_{time}$-R relations and J $m^{-2}$ $mm^{-1}$ for $KE_{mm}$-R relations and the same "Note: Units for RMSE are J $m^{-2}$ $h^{-1}$ for $KE_{time}$-R relations and J $m^{-2}$ $mm^{-1}$ for $KE_{mm}$-R relations" is provided in table (Table #3) caption of revised manuscript at lines 705-706.

**Response to Referee # 03** (Report #01)

**GENERAL COMMENTS**

The manuscript mainly analyzed RSD data collected in north Taiwan during Typhoon and non-typhoon events. Furthermore, a brief analysis of additional data (reanalysis, remote-sensing, and ground-based datasets) is also provided. The manuscript is well written and easy to follow. I suggest the publication on Hydrology and Earth System Sciences after addressing my minor comments.

**Reply:** We sincerely acknowledge the reviewer for providing encouraging and precise comments. We have addressed all the reviewer's comments by providing detailed response for each comment, and the necessary modifications are made (with red color text) in the revised manuscript.

**SPECIFIC COMMENTS:**

**1.** Lines 192-194: the sentence need to be further explained. Please add some details

**Reply:** As per the reviewer's suggestion, the sentence is modified as: "Further, a statistical Student's t-test (used to determine whether two data sets are significantly different from each other or not), is executed between TY and NTY rainfall $D_m$ values. The test results rejected the null hypothesis at 0.05 and 0.01 significance levels, which confirm that the $D_m$ values in TY rainfall are different from that of the NTY rainfall. Similarly, the Student's t-test performed for other three parameters ($\log_{10}N_w$, $\log_{10}R$, and $\log_{10}W$) also showed that these parameters in TY rainfall are different from that of the NTY rainfall." in the revised manuscript at line 212-218.

**2.** Figure 5: check the y-label of Figure 5c & d

**Reply:** We thank the reviewer for this comment. The y-label of Fig. 5c & d are changed to "Percentage of $N_t$ (m$^{-3}$)" and "Percentage of $R$ (mm h$^{-1}$)" and the modified figure is provided in the revised manuscript.

**3.** Lines 221-224: Please explain better the separation criterion for typhoon and non-typhoon events, so that it can be easily applied also to other researchers.

**Reply:** We thank the reviewer for this comment. The sentence in lines 221-224 describes the criteria for considering each rainfall rate class (C1: $0.1 \leq R < 1$, C2: $1 \leq R < 2$, C3: $2 \leq R < 5$, C4: $5 \leq R < 10$, C5: $10 \leq R < 30$, C6: $30 \leq R < 50$, and C7: $R > 5$, where $R$ is in mm h$^{-1}$; please refer to table 1), but not the "separation criterion for typhoon and non-typhoon events".

The typhoon (TY) and non-typhoon (NTY) rainfall events separation criteria is provided at lines: 115-118, i.e., During summer seasons, if there is any typhoon track within 500 km radius from the disdrometer site, the corresponding rainfall is considered as typhoon (TY)

rain event, and the rest of the rainfall events in summer seasons are considered as non-typhoon (NTY) rain events.

**4.** Lines 231-232: please write the equation! It will be more clear to the reader

**Reply:** As per the reviewer's recommendations, Equation is provided for the percentage parameters as given below in the revised manuscript at lines 258-265.

"The percentage parameter of $N(D)$ for different rain rate class, $\delta(D, R) = \delta(D, R_{Ck})_{TY/NTY}$ is given as

$$\delta(D, R_{Ck})_{TY} = \frac{[N(D)_{TY}]_{Ck}}{([N(D)_{TY}]_{Ck} + [N(D)_{NTY}]_{Ck})} \times 100 \qquad \text{----------(1)}$$

$$\delta(D, R_{Ck})_{NTY} = \frac{[N(D)_{NTY}]_{Ck}}{([N(D)_{TY}]_{Ck} + [N(D)_{NTY}]_{Ck})} \times 100 \qquad \text{----------(2)}$$

Where $[N(D)_{TY}]_{Ck}$ or $[N(D)_{NTY}]_{Ck}$ represents the mean $N(D)$ of TY or NTY rainfall for the rain rate class "Ck", with k=1, 2, 3, 4, 5, 6, 7 (C1: $0.1 \leq R < 1$, C2: $1 \leq R < 2$, C3: $2 \leq R < 5$, C4: $5 \leq R < 10$, C5: $10 \leq R < 30$, C6: $30 \leq R < 50$, and C7: $R > 50$, where $R$ is in mm h$^{-1}$; please refer to table 1)."

**5.** Line 251: Why the Authors did not use the C1-C7 rain classes in Figure 8 as in Figure 6 and Figure 7?

**Reply:** To have a consistency among Fig. 6, Fig. 7 and Fig.8, the seven rainfall rate classes (C1-C7) are used in Fig.8 and the modified figure is provided in the revised manuscript.

**6.** Lines 274-276: Why the Authors did not use the Bringi et al. method described and analysed in the previous section? Please use that method or eliminate it and include Me et al. method in the analysis reported in the previous section of the manuscript or justify why you analysed your data with respect to the Bringi et al. method and then you used the Ma et al method for classification.

**Reply:** We thank the reviewer for this meticulous comment.

The main purpose plotting Fig. 8 is to know how $D_o$ (mm), and $\log_{10}N_w$ ($N_w$ in m$^{-3}$ mm$^{-1}$) distributions vary at different rainfall rate classes for both TY and NTY rainfall and then to notice the performance of Bringi et al. (2003) and Thompson et al. (2015) rain classification methods. From Fig.8 we noticed that in classifying the TY and NTY rainfall into stratiform and convective type, Bringi et al. (2003) classification method is superior to that of the Thomson et al. (2015).

In Bringi et al. (2003) rain classification, 5 consecutive 2-min (or 10 consecutive 1-min) RSD samples are considered to as stratiform type if the mean value of rainfall rate ($R$) $\geq 0.5$ mm h$^{-1}$ and the standard deviation of $R$ ($\sigma_R$) $\leq 1.5$ mm h$^{-1}$, and convective type if the mean value of $R > 5$ mm h$^{-1}$ and the standard deviation of $R$ ($\sigma_R$) $> 1.5$ mm h$^{-1}$. However, with this method, we may miss some RSD samples that can't satisfy the above two conditions.

On the other hand, the Ma et al. (2019) rain classification method is the modified form of Bringi et al. (2003) ["*if the standard derivation of rain rate for a consequent 10 min is greater than 1.5 mm h$^{-1}$ and the rain rate is greater than 5 mm h$^{-1}$, it is classified as convective rain; otherwise, it is classified as stratiform rain."* : from Page # 4157, Section 3.2 of Ma et al. (2019) ].

As there is possibility for losing some RSD samples while classifying the precipitation into stratiform and convective type using Bring et al. (2003) method, to accommodate all RSD samples into stratiform or convective category, we adopted the modified form of Bringi et al. (2003) rain classification procedure as mention in Ma et al. (2019), i.e., "*if the mean value of rain rate for a consequent 10 min is > 5 mm h$^{-1}$and the standard derivation of rain rate is > 1.5 mm h$^{-1}$, it is classified as convective rain; otherwise, it is classified as stratiform rain"*

Hence, the sentence is modified as given below in the revised manuscript at lines 304-306.

"In separating the TY and NTY rainfall into stratiform and convective type, we adopted the modified form of Bring et al. (2003) classification method as mentioned in Ma et al. (2019)."

**7.** Line 298: "is due the presence" instead of "is due the presence"
**Reply:** The typo error is corrected by replacing "is due the presence" with "is due to the presence" in the revised manuscript at line 329.

**8.** Line 303: please see Adirosi et al. (2018) for the effects of different disdrometer types on the Z-R relation.
Adirosi, E., Roberto, N., Montopoli, M., Gorgucci, E., & Baldini, L. (2018). Influence of disdrometer type on weather radar algorithms from measured DSD: Application to Italian climatology. Atmosphere, 9(9), 360.
**Reply:** We thank the reviewer for suggesting this article that provides the information about how different types of disdrometers influences the weather radar algorithms. We have gone through this article and we mentioned it the revised manuscript at line 334.

**9.** Line 357: please specify the meaning of CER
   **Reply:** We have mentioned the meaning of CER as "cloud effective radii (CER)" in the revised manuscript at line 173.

---

## Author Response (AR3)

**Response to Referee # 03** (Report #02)

Review of hess-2020-345-manuscript-version4

By Jayalakshmi Janapati, Balaji Kumar Seela, Pay-Liam Lin, Meng-Tze Lee , Everette Joseph

Manuscript Title ‑ Microphysical features of typhoon and non-typhoon rainfall observed in Taiwan, an island in the northwest Pacific

**GENERAL COMMENTS**

Authors addressed all my comments. I have only one minor comment.

**SPECIFIC COMMENTS:**

1) If I understood well, in the answer to my previous comment n6, the one regarding the use of Bringi et al. methods to separate between stratiform and convective rain, the Authors wrote that in Figure 8 of the paper they use Bringi et al. (2003) classification line as reference and then in the analysis they used the Ma et al. (2019) method that is derived from the Bringi et al. (2003) method. Please note that in the text in section 3.2 and in the caption of Figure 8 the Authors refer to Bringi et al. (2009) and not to Bringi et al. (2003). Furthermore, I understand the need of using Ma et al. (2019) instead of Bringi et al. (2003), but probably, for consistency the line obtained with Ma et al. (2019) can be added to Figure 8, to understand its performance with respect to Bringi et al. (2003).

   [ *Previous comment # 5: Lines 274-276: Why the Authors did not use the Bringi et al. method described and analysed in the previous section? Please use that method or eliminate it and include Me et al. method in the analysis reported in the previous section of the manuscript or justify why you analysed your data with respect to the Bringi et al. method and then you used the Ma et al method for classification.]*

**Reply:** We thank the reviewer for this comment.
To provide the consistency and ease of understating to the readers, as per the reviewer's previous recommendation, we have eliminated Figure 8 and its related text in the revised manuscript.